**Investigation of spatial and temporal variability in lower tropospheric ozone**
**from RAL Space UV-Vis satellite products**
Richard J. Pope[1,2], Brian J. Kerridge[3,4], Richard Siddans[3,4], Barry G. Latter[3,4], Martyn P. Chipperfield[1,2], Wuhu
Feng[1,5], Matilda A. Pimlott[1], Sandip S. Dhomse[1,2], Christian Retscher[6] and Richard Rigby[1,7]
*1: School of Earth and Environment, University of Leeds, Leeds, United Kingdom*
*2: National Centre for Earth Observation, University of Leeds, Leeds, United Kingdom*
*3: Remote Sensing Group, STFC Rutherford Appleton Laboratory, Chilton, United Kingdom*
*4: National Centre for Earth Observation, STFC Rutherford Appleton Laboratory, Chilton, United Kingdom*
*5: National Centre for Atmospheric Science, University of Leeds, Leeds, United Kingdom*
*6: European Space Agency, ESRIN, Frascati, Italy*
*7: Centre for Environmental Modelling and Computation, University of Leeds, Leeds, United Kingdom*
Revised version for *Atmospheric Chemistry and Physics*
*Correspondence to*: Richard J. Pope (r.j.pope@leeds.ac.uk)
**Key Points**
- The RAL Space profile retrieval algorithm for ultraviolet-visible nadir sounders has good vertical
sensitivity to retrieve lower tropospheric column ozone ($LTCO_3$).
- OMI, SCIAMACHY and GOME-1 have suitably stable $LTCO_3$ records in comparison to ozonesondes
and are merged to form the first long-term satellite $LTCO_3$ record (1996-2017).
- Comparison of 5-year averages for 1996-2000 and 2013-2017 suggests a significant $LTCO_3$ increase
(3.0 to 5.0 DU) in the tropics/sub-tropics over the satellite-era.
**Abstract:**
Ozone is a potent air pollutant in the lower troposphere and an important short-lived climate forcer (SLCF) in
the upper troposphere. Studies using satellite data to investigate spatiotemporal variability of troposphere
ozone ($TO_3$) have predominantly focussed on the tropospheric column metric. This is the first study to
investigate long-term spatiotemporal variability in lower tropospheric column ozone ($LTCO_3$, surface-450 hPa
sub-column) by merging multiple European Space Agency – Climate Change Initiative (ESA-CCI) products
produced by the Rutherford Appleton Laboratory (RAL) Space. We find that in the $LTCO_3$, the degrees of
freedom of signal (DOFS) from these products varies with latitude range and season and is up to 0.8,
indicating that the retrievals contain useful information on lower $TO_3$. The spatial and seasonal variation of
the RAL Space products are in good agreement with each other but there are systematic offsets of up to 3.0-
5.0 DU between them. Comparison with ozonesondes shows that the Global Ozone Monitoring Experiment
(GOME-1, 1996-2003), the SCanning Imaging Absorption spectroMeter for Atmospheric
CartograpHY (SCIAMACHY, 2003-2010) and the Ozone Monitoring Instrument (OMI, 2005-2017) have stable
$LTCO_3$ records over their respective periods, which can be merged together. However, GOME-2 (2008-2018)
shows substantial drift in its bias with respect to ozonesondes. We have therefore constructed a robust
merged dataset of $LTCO_3$ from GOME-1, SCIAMACHY and OMI between 1996 and 2017. Comparing the
$LTCO_3$ differences between the 1996-2000 and 2013-2017 5-year averages, we find sizeable positive
increases (3.0-5.0 DU) in the tropics/sub-tropics, while in the northern mid-latitudes, we find small scale
differences in $LTCO_3$. Therefore, we conclude that there has been a substantial increase in tropical/sub-
tropical LTCO$_3$ during the satellite-era, which is consistent with tropospheric column ozone (TCO$_3$) records
from overlapping time-periods (e.g. 2005-2016).

## 1. Introduction

Tropospheric ozone (TO$_3$) is a short-lived climate forcer (SLCF) and, is the third most important greenhouse
gas (GHG; e. g. Myhre et al., 2013). TO$_3$ is also a hazardous air pollutant with adverse impacts on human
health (WHO, 2018) and the biosphere (e.g. agricultural and natural vegetation; Sitch et al., 2007). Since the
pre-industrial (PI) period, anthropogenic activities have increased the atmospheric loading of ozone (O$_3$)
precursor gases, most notably nitrogen oxides (NO$_x$) and methane (CH$_4$), resulting in a substantial increase in
TO$_3$ of 25-50% since 1900 (Gauss et al., 2006; Lamarque et al., 2010; Young et al., 2013). The PI to present
day (PD) radiative forcing (RF) from TO$_3$ is estimated by the Intergovernmental Panel on Climate Change
(IPCC) to be 0.47 Wm$^{-2}$ (Forster et al., 2021) with an uncertainty range of 0.24-0.70 Wm$^{-2}$.
During the satellite-era, with a number of missions since 2000, extensive records of TO$_3$ have been
produced, e.g. by the European Space Agency Climate Change Initiative (ESA-CCI; ESA, 2019). However, the
large overburden of stratospheric O$_3$, coupled with the different vertical sensitivities and sources of error
associated with observations in different wavelength regions (e.g. Eskes and Boersma 2003; Ziemke et al.,
2011; Miles et al., 2015) contributes to large-scale spatiotemporal inconsistencies between the records
(Gaudel et al., 2018). Various studies (e.g. Heue et al. 2016; Pope et al., 2018; Ziemke et al. 2019) analysing
TO$_3$ trends usually focussed on one or two instruments. The work by Gaudel et al. (2018) was part of the
Tropospheric Ozone Assessment Report (TOAR), which represented a large global effort to understand
spatiotemporal patterns and variability in TO$_3$. Gaudel et al., (2018) analysed ozonesondes and multiple
polar orbiting-nadir viewing satellite products and reported large-scale discrepancies in the spatial
distribution, magnitude, direction and significance of the TCO$_3$ trends. While the satellite records did cover
slightly different time periods, they were unable to provide any definitive reasons for these discrepancies
beyond briefly suggesting that differences in measurement techniques and retrieval methods were likely to
be causing the observed spatial inconsistencies. Another factor introducing inconsistencies is the assumed
tropopause height for the different products. Some products used the World Meteorological Organisation
(WMO) definition of "the first occurrence of the 2 K/km lapse-rate" while some others e.g. integrated the 0-
6 km and 6-12 km sub-columns to derive the tropospheric column. The use of different a priori products
within the retrieval scheme will have also provided inconsistencies.
The vertical sensitivity of each product (function of measurement technique and retrieval methodology)
used by Gaudel et al. (2018) has a substantial impact on which part of the troposphere (and stratosphere)
the O$_3$ signal is weighted towards. The vertical sensitivity can be referred to as the "averaging kernel" (AK),
which provides the relationship between perturbations at different levels in the retrieved and true profiles
(Rodgers, 2000; Eskes and Boersma, 2003). As the instruments' vertical sensitivities differ, they are likely to
be influenced differently by processes controlling TO$_3$ temporal variability in different layers of the
troposphere (e.g. lower troposphere influenced more by precursor emissions vs. the upper troposphere
subject more to the influence from stratospheric-tropospheric exchange). Therefore, the differing vertical
sensitivities, and thus the TO$_3$ they are retrieving, could be driving the inconsistencies in reported TCO$_3$
trends between products.
While many studies have previously focussed on TCO$_3$ (e.g. Gaudel et al. (2018); Ziemke et al. (2019)),
several nadir-viewing ultraviolet-visible (UV-Vis) sounders can retrieve TO$_3$ between the surface to 450 hPa
(i.e. lower tropospheric column O$_3$, LTCO$_3$). The retrieval scheme from the Rutherford Appleton Laboratory
(RAL) Space exploits information from the O$_3$ Huggins bands (325-335 nm), as well as the Hartley band (270-
307nm), to retrieve high quality $LTCO_3$ and was selected for the ESA-CCI and EU Copernicus Climate Change
Service. As a result, the RAL Space $LTCO_3$ products (and equivalent from other providers) are valuable
resources to investigate global and regional $O_3$-related air quality (e.g. Richards et al., 2013; Pope et al.,
2018; Russo et al., 2023).
In this study, we explore the spatiotemporal variability of $LTCO_3$ from several UV-Vis sounders produced by
RAL Space. While Gaudel et al., (2018) used a range of UV-Vis and infrared (IR) $TCO_3$ products, including the
RAL Space Ozone Monitoring Instrument (OMI) product, we focus here on several RAL Space UV-Vis
products. Here, we aim to explore the consistencies between them, their vertical sensitivities, $LTCO_3$ stability
against ozonesonde records and suitability for long-term trend analysis. In our manuscript, Section 2
discusses the satellite/ozonesonde datasets used, Section 3 presents are results, while Section 4 summarises
our conclusions and discussion points.
**2.   Methodology and Datasets**
**2.1. Datasets**
The four RAL Space UV-Vis satellite products investigated here are from OMI, the Global Ozone Monitoring
Experiment – 1 (GOME-1), GOME-2 and the SCanning Imaging Absorption spectroMeter for Atmospheric
CartograpHY (SCIAMACHY), all of which were developed as part of the ESA-CCI project (**Table 1**). GOME-1,
GOME-2, SCIAMACHY and OMI flew on ESA's ERS-2, MetOp-A, ENVISAT and NASA's Aura satellites in sun-
synchronous low Earth polar orbits with local overpass times of 10.30, 9.30, 10.00 and 13.30, respectively.
They are all nadir viewing with spectral ranges which include the 270-350 nm range used for ozone profile
retrieval. The spatial footprints of the respective instruments at nadir are 320 km × 40 km, 80 km × 40 km,
240 km × 30 km and 24 km × 13 km (Boersma et al., 2011; Miles et al., 2015; Shah et al., 2018). The scheme
established by RAL Space to retrieve height-resolved $O_3$ profiles with tropospheric sensitivity (Miles et al.,
2015) was applied to all of these satellite instruments. The scheme is based on the optimal estimation (OE)
approach of Rogers et al., (2000) and provides state-of-the-art retrieval sensitivity to lower $TO_3$, which is
described in detail by Miles et al., (2015) and by Keppens et al., (2018). The differences between the retrieval
versions (i.e. fv214 and fv300) in **Table 1** are primarily linked to the instrument types where GOME-1, GOME-
2 and SCIAMACHY are across-track scanning instruments while OMI uses a 2-D array detector. For this work,
the data were filtered for good quality retrievals whereby the geometric cloud fraction was <0.2, the lowest
sub-column $O_3$ value was > 0.0, the solar zenith angle < 80.0°, the convergence flag = 1.0 and the normalised
cost function was < 2.0. These filters also remove OMI pixels influenced by the OMI row anomaly (Torres et
al., 2018), so there is reduced OMI data coverage over the record. However, we find this has minimal impact
on our results with substantial proportions of data (e.g. millions of retrievals per year at the start and end of
the OMI record) available for analysis in our study.
**2.2. Ozonesondes and Application of Satellite Averaging Kernels**
To help understand the impact of the satellite AKs on retrieved $LTCO_3$ and stability of the satellite
instruments listed in **Table 1** over time, we use ozonesonde data between 1995 and 2019 from the World
Ozone and Ultraviolet Radiation Data Centre (WOUDC), the Southern Hemisphere ADditional Ozonesondes
(SHADOZ) project and from the National Oceanic and Atmospheric Administration (NOAA). Keppens et al.,
(2018) undertook a detailed assessment of the ESA-CCI $TO_3$ data sets, including the RAL UV-Vis profile data
sets used in this study (mostly older versions though) using ozonesondes. They found that the RAL $LTCO_3$
products typically had a positive bias of about 40%, apart from OMI which was closer to 10%. On the global
scale, tropospheric drift in GOME-1 and OMI over time was approximately -5% and 10% per decade,
respectively. However, GOME-2 and SCIAMACHY had significant tropospheric drift trends of approximately
40% per decade. The recent Copernicus *Product Quality Assessment Report (PQAR) Ozone Products Version*
*2.0b* (Copernicus, 2021) undertook a more recent assessment of nadir ozone profiles using the level 3
products from RAL listed in **Table 1.** They found that in the troposphere, OMI/GOME-1 and
SCIMACHY/GOME-2 had biases of -20% and 10%. GOME-1 tropospheric drift was deemed to be insignificant
(-10% to 5% per decade), while GOME-2 and SCIAMACHY had a significant drift of 30% and 20% per decade,
respectively. OMI also had an insignificant tropospheric drift of 10% per decade.
In this study, for comparisons between ozonesonde profiles and satellite retrievals, each ozonesonde profile
was spatiotemporally co-located to the closest satellite retrieval. Here, all the retrievals within 6 hours of the
ozonesonde launch were subsampled and then the closest retrieval in space (i.e. within 500 km) was taken
for the final co-located one. Therefore, there was one satellite retrieval for every ozonesonde profile to help
reduce the spatiotemporal sampling difference errors. Here, ozonesonde $O_3$ measurements were rejected if
the $O_3$ or pressure values were unphysical (i.e. < 0.0), if the $O_3$ partial pressure > 2000.0 mPa or the $O_3$ value
was set to 99.9, and whole ozonesonde profiles were rejected if at least 50% of the measurements did not
meet these criteria. These criteria are similar to those applied by Keppens et al., (2018) and Hubert et al.,
(2016). To allow for direct like-for-like comparisons between the two quantities, accounting for the vertical
sensitivity of the satellite, the instrument AKs were applied the ozonesonde profiles. Here, each co-located
ozonesonde profile (in volume mixing ratio) was used to derive ozone sub-columns (in number density) on
the satellite pressure grid. The application of the AKs for the UV-Vis instruments was done using **Equation 1**:
$$sonde_{AK} = AK.(sonde_{int} - apr) + apr \qquad (1)$$
where $sonde_{AK}$ is the modified ozonesonde sub-column profile (Dobson units, DU), $AK$ is the averaging kernel
matrix, $sonde_{int}$ is the sonde sub-column profile (DU) on the satellite pressure grid and $apr$ is the apriori sub-
column amount (DU). Here, the ozonesonde profile, on its original pressure grid (typically in units of ppbv or
mPa) are converted into ozone sub-columns between each pair of measurement levels. These sub-columns
are then aggregated up to the larger sub-columns (e.g. the $LTCO_3$ range is between the surface and 450 hPa)
on the coarser satellite pressure grid.

## 3. Results
### 3.1. Satellite Vertical Sensitivity
**Figure 1** represents average AKs for all the instruments listed in **Table 1** for 2008 (1998 for GOME-1) in the
northern (NH) and southern (SH) hemispheres between the equator and 60°S & N. Of the four RAL Space
products, OMI $O_3$ profiles appear to contain the most information with degrees of freedom of signal (DOFS)
of 5.0 or above for the full atmosphere. Here, the DOFS represents the number of independent pieces of
information on the vertical profile in the retrieval (i.e. the sum of the AK diagonal). SCIAMACHY has the
lowest sensitivity with average DOFS ranging between 4.12 and 4.64. The DOFS tends to be larger in NH for
all the products, though there is no clear pattern in the seasonality (i.e. January vs. July). In terms of $LTCO_3$,
OMI again has greater sensitivity than the others with average hemispheric and seasonal DOFS ranging
between 0.63 and 0.68. For GOME-1 (GOME-2), the $LTCO_3$ DOFS range between 0.37 and 0.50 (0.39 and
0.46). SCIAMACHY $LTCO_3$ DOFS range between 0.44 and 0.52. Therefore, while SCIAMACHY has the lowest
overall information on the full atmospheric ozone, it has reasonably good information in the $LTCO_3$, as do
the other instruments. These results are robust given the large number of retrievals (N) that have been used
to derive the average AKs (i.e. N > 65,000 in all cases).
While **Figure 1** provides spatial average information on $LTCO_3$ DOFS, **Figure 2** shows spatial maps for
December-January-February (DJF) and June-July-August (JJA) over the respective instrument records. The
largest $LTCO_3$ DOFs occur over the ocean ranging between approximately 0.4 and 0.6 for GOME-1, GOME-2
and SCIAMACHY, while OMI has larger ocean values between 0.7 and 0.8. Over land, the $LTCO_3$ DOFS tend to
be lower and between 0.3 and 0.5 for GOME-1, GOME-2 and SCIAMACHY. Again, OMI has larger values on
land of between 0.4 and 0.7. Depending on the hemispheric season, the summer-time (JJA in NH and DJF in
SH) LTCO$_3$ DOFS are larger for each instrument. Overall, OMI (GOME-2) retrievals contain the largest (lowest)
amount of information on LTCO$_3$.
The impact of the satellite vertical sensitivity is further investigated by co-locating the products with the
merged ozonesonde data set, over their respective mission periods (globally and in the NH and SH) and the
AKs applied to assess the impact on the ozonesondes (**Figure 3**). For all the instruments, there are suitable
samples sizes (N > 1000 in all cases) of co-located retrievals and derived ozonesonde LTCO$_3$. In the case of
GOME-1, the global distribution has a 25$^{th}$-75$^{th}$ percentile (25_75%) range of approximately 8.0 to 20.0 DU
and a median of 14.0 DU. The apriori 25_75% range and median values are 16.0 to 22.0 and 19.0 DU. These
substantial differences between retrieved and apriori values confirm there is sensitivity in the GOME-1
retrieval to lower tropospheric ozone. It can be seen from **Equation 1** that if a satellite instrument had
perfect sensitivity at all levels (i.e. AK=1), there would be no change in co-located ozonesonde LTCO$_3$
distribution when the AKs are applied. However, given AK values are less than 1.0 in **Figure 1**, leading to the
DOFS of approximately 0.5, there is a shift in the median value towards the apriori from approximately 21.0
to 19.0 DU. The corresponding ozonesonde 10$^{th}$-90$^{th}$ percentile (10_90%) range of 13.0 to 26.0 DU expanded
to 12.0 to 27.0 DU. Therefore, the application of the AKs to the ozonesondes actually increases the range of
observed values. In the NH, the GOME-1 median (25_75% range) is 14.0 (4.0-24) DU while the apriori median
(25_75% range) is 21.0 (18.0-23.0) DU. The ozonesonde median (25_75% range) is 22.0 (19.0-25.0) DU while
application of the AKs yields values of 19.0 (16.0-24.0) DU. In the SH, the GOME-1 median (25_75% range) is
12.0 (8.0-17.0) DU while the apriori median (25_75% range) is 14.0 (12.0-16.0) DU. The ozonesonde median
(25_75% range) is 12.0 (11.0-17.0) DU while application of the AKs yields values of 12.0 (6.0->40.0) DU. In
comparison, GOME-2 shows a similar response though the shift in LTCO$_3$ value between the apriori and
satellite is smaller. This makes sense given the lower vertical sensitivity of GOME-2. In the SH, the application
of the AKs to the ozonesondes yields a very large range in the percentiles. It is likely that the South Atlantic
Anomaly (SAA – i.e. where charged particles directly impact UV detectors increasing dark-current noise,
which in turn reduces the number of retrievals from all UV sensors, notably both GOME-1 and GOME-2;
Keppens et al., 2018), given the typically larger values and signal corruption, is driving the large response in
the ozonesonde+AKs range.
For OMI, the global distribution has a median (25_75% range) of 17.0 (13.0-25.0) DU yielding a substantial
shift from the apriori median (25_75% range) of 18.0 (16.0-22.0) DU. In the NH, the satellite median (25_75%
range) is 18.0 (13.0-25.0) DU and the apriori median (25_75% range) value is 20.0 (17.0-23.0) DU. In the SH,
the satellite median (25_75% range) is 14.0 (10.0-22.0) DU and the apriori median (25_75% range) value of
15.0 (13.0-19.0) DU. When the AKs are applied to the ozonesondes there is typically an increase in the
median LTCO$_3$ and range by approximately 3.0-4.0 DU. This increase in LTCO$_3$ when the OMI AKs are applied
to the ozonesondes contrasts with the other satellite instruments. While the vertical smearing from the
stratosphere would intuitively be expected to increase the tropospheric layer retrieval, and thus the AK
adjustment to decrease the ozonesonde value, in the case of OMI there is a negative excursion in the AKs
into the lowermost stratosphere (see **Figure 1**), so the opposite occurs. For SCIAMACHY, a similar
relationship occurs to that of GOME-1 and GOME-2 with a shift of the satellite LTCO$_3$ median away from the
apriori by 1.0-3.0 DU and an increase the in 25_75% range by 10.0-15.0 DU. Apart from the SH, the
application of the AKs to the ozonesondes shifts the LTCO$_3$ median by 2.0-3.0 DU but the 25_75% range
remains similar. Overall, there is shift in the satellite LTCO$_3$ median value away from the apriori with an
increase in the 25_75% and 10_90% ranges. A similar pattern occurs in multiple cases between the
ozonesondes and the ozonesondes+AKs. Therefore, all the instruments have reasonable vertical sensitivity in
LTCO$_3$ with substantial perturbations from the apriori and to the satellite LTCO$_3$ distribution.

### 3.2. Lower Tropospheric Column Ozone Seasonality

Multiple studies have investigated the seasonality of TO$_3$ from space observing large biomass burning and
lightning induced O$_3$ in the South Atlantic (Ziemke et al., 2006; Ziemke et al., 2011; Pope et al., 2020),
enhanced summertime TO$_3$ over the Mediterranean (Richards et al., 2013), TO$_3$ over large precursor regions
such as China and India (Verstraeten et al., 2015) and the enriched northern hemispheric background O$_3$
during springtime (Ziemke et al., 2006). Here, we compare the long-term seasonal (DJF and JJA) spatial
distributions of RAL Space LTCO$_3$ products (**Figure 4**).
OMI and GOME-2 LTCO$_3$ have regions of consistency (e.g. JJA NH enhanced background TO$_3$, between 20.0
DU and 30.0 DU, and the Mediterranean TO$_3$ peak, >25.0 DU), but the SAA interferes with the signal of the
biomass burning induced secondary O$_3$ formation from Africa and South America. However, for OMI, this
ozone plume ranges between 23.0 and 27.0 DU (18.0 and 20.0 DU) in DJF (JJA). There are also clear LTCO$_3$
hotspots over anthropogenic regions (e.g. eastern China and northern India) peaking at over 25.0 DU in JJA.
The GOME-1 LTCO$_3$ spatial patterns are consistent with that of OMI and GOME-2, but there is a systematic
low bias relative to OMI and GOME-2 in the absolute LTCO$_3$ of 3.0 DU to 7.0 DU, depending on geographical
location (e.g. 20.0-22.0 DU over northern India for GOME-2 and OMI, while 16-18 DU for GOME-1). These
differences in the GOME-1 and GOME-2/OMI LTCO$_3$ seasonal averages are likely to be at least partly due to
underlying LTCO$_3$ tendencies between the respective instrument time periods. This is investigated further in
Section 3.4. The SCIAMACHY spatial pattern and absolute LTCO$_3$ values are more consistent with OMI and
GOME-2. Moreover, SCIAMACHY shows limited sensitivity to the SAA and resolves the biomass burning /
lightning O$_3$ sources detected by OMI over South America, South Atlantic and Africa (18.0-20.0 DU in JJA).
However, especially in the NH in DJF, there appears to be regions of latitudinal banding in the LTCO$_3$ spatial
patterns (e.g. 0°-30°N), which are not observed (or to the same extent) as the other UV-Vis sounders.
Overall, GOME-2 and OMI are in good agreement spatially and seasonally with similar absolute LTCO$_3$ values.
In DJF and JJA, OMI appears to be 2.0-3.0 DU lower and larger than GOME-2, respectively. This is reasonable
given the similar temporal records they cover (2005-2017 vs. 2007-2018). SCIAMACHY has similar spatial-
seasonal patterns but has systematically larger (3.0-5.0 DU) DJF values in comparisons to OMI and GOME-2.
The satellite LTCO$_3$ seasonality is consistent with that of the ozonesondes. Here, the median (25[th] percentile,
75[th] percentile) ozonesonde LTCO$_3$ values for the NH in DJF, NH in JJA, SH in DJF and SH in JJA are 18.0 (15.7,
20.0) DU, 20.8 (16.7, 24.6) DU, 10.8 (8.2, 14.8) DU and 14.4 (12.1, 16.3) DU, respectively. Therefore, the NH
LTCO$_3$ values are larger than those in the SH and the JJA LTCO$_3$ values are larger than the DJF equivalent. All
of which are consistent with the four instrument LTCO$_3$ seasonal distributions.

### 3.3. Satellite Instrument Temporal Stability

For accurate assessment of satellite LCTO$_3$ temporal variability, there needs to be insignificant drift over
time, whereas bias which is constant over time can be tolerated. The most appropriate data set with which
to assess satellite long-term drifts is that of the ozonesonde record, albeit that it has certain limitations
potentially including temporal changes in accuracy (Stauffer et al., 2020) as well as geographical coverage.
**Figure 5** shows annual time series of the satellite-ozonesonde (with AKs applied) median biases for three
latitude bands: 90°-30°S, 30°S-30°N and 30-90°N. The hatched pixels show where the biases are non-
substantial, defined as the 25_75% difference range intersecting with zero. For GOME-1, the mean bias (MB)
is -5.34, -3.21 and -0.90 DU for the three regions, respectively. For the 30-90°N region, several years show
substantial biases of -6.0 to -3.0 DU. The two other latitude bands have few substantial years but in the
tropical band, both 2002 and 2003 show substantial biases of approximately -5.0 DU. To assess the stability
of the instruments with time, a simple linear least-squares fit was performed with regional trends of -0.32, -
0.98* and -0.03 DU/yr. A substantial trend (shown by an asterisk) has a p-value < 0.05 as defined as
$|M/\sigma_M| > 2.0$ (e.g. Pope et al., 2018), where $M$ and $\sigma_M$ are the linear trend and trend uncertainty,
respectively. While, the 30-90°N region had a sizable systematic bias, it was stable with time, as was the bias
for the 90°-30°S region. However, the 2002 and 2003 biases in the 30S°-30°N region gave rise to a substantial
drift in the GOME-1 record.
For GOME-2, the record MB is 1.91, -5.05 and 1.64 DU for the respective latitude bands, all of which have
substantial bias trends at 0.62*, -0.70* and 0.22* DU/yr. Therefore, the GOME-2 LTCO$_3$ records from this
processing run are not stable and cannot be used further in the study. SCIAMACHY has regional mean biases
of 1.33, 4.47 and 2.81 DU. In the 30-90°N region, the bias is non-substantial. While there are substantial
biases peaking at 3.0-5.0 DU in the 90°-30°S region, neither region has a critical drift trend. The largest
substantial biases are in the 30S°-30°N region (>5.0 DU) for 2006 to 2008. While the positive trend of 0.21
DU/yr is insignificant, we do not use the SCIAMACHY data in later years when harmonising the LTCO$_3$ records
(section 3.4). OMI has MBs of -5.16, -2.91 and -0.41 DU with only a few of the year-latitude pixels having
substantial biases peaking at -6.0 to -3.0 DU in the 30-90°N region. The resulting bias trends are -0.12, 0.22
and -0.10 DU/yr, which all have p-values > 0.05. Therefore, GOME-1, OMI and SCIAMACHY were deemed
suitable LTCO$_3$ records for use in this study.
**3.4. Lower Tropospheric Column Ozone Merged Record**
The RAL Space products cover the full period between 1996 and 2017. Therefore, there is the opportunity to
merge and harmonise these records to produce a long-term record to look at the spatiotemporal variability
of LTCO$_3$. From **Figure 5,** the OMI record appears to be stable with time globally, providing a suitable data set
between 2005 and 2017. The GOME-2 record appears not to be sufficiently stable across its record (2008-
2018), so is not included in subsequent analysis. The GOME-1 record covers 1996 to 2010, but given the loss
of geographical coverage due to the onboard tape recorder failing in June 2003 (van Roozendael, 2012), a
true global average is only available between 1996 and 2003. **Figure 5** shows that GOME-1 bias with respect
to the ozonesonde record is not stable in the tropics but this is predominantly driven by instrument-
ozonesonde differences in 2003. Therefore, 2003 is also dropped leaving the GOME-1 global record between
1996 and 2002. The GOME-1 tropical bias for 2002 is similar to that of 2003 (-5.0 DU) but the biases for the
other latitude bands are less distinct. The regional average LTCO$_3$ values for 2002 in **Figure 5** are also
comparable to neighbouring years (e.g. 2000 and 2001). SCIAMACHY also does not have a full year of data
for 2002, so we have included the GOME-1 2002 data in our analysis.
While OMI (2005-2017) and GOME-1 (1996-2002) now cover a large proportion of the global record, there is
still a systematic difference between them. Different UV-Vis instruments can have inconsistencies in their
retrieved products (e.g. van der A et al., (2006), Heue et al., (2016)) and often require a systematic
adjustment to create a harmonised record. Here, there is overlap in the raw records between 2005 and 2010
for GOME-1 and OMI. The GOME-1 record does have large missing data gaps globally, but for the mid-
latitude and tropical latitude bands, there is sufficient sampling to inter-compare the two records. Therefore,
for each swath, the nearest OMI retrieval is co-located to that of GOME-1, but has to be within 250 km. The
local overpass times are different (i.e. GOME-1 10.30 and OMI 13.30) but within approximately 3-hours, so
the diurnal cycle impacts are likely to be of a secondary order and we are confident in merging the records.
Based on the co-located OMI and GOME-1 data, we derived long-term latitude-month offset which are
added to GOME-1 (1996-2002) to harmonise the records. The was done using latitudinal bins of 60°S-30°S,
30°S-30°N and 30°N-60°N. Given the lack of GOME-1 data outside of 60°S-60°N due to the failure of the
GOME-1 tape recorder in June 2003, there was insufficient data to derive offsets, the high-latitudes data is
excluded in the following sections. Where there was good spatial coverage from GOME-1 between 2005 and
2010, once the offset had been applied, gridded OMI and GOME-1 where data existed for both, on a pixel by
pixel basis, were averaged together.
For 2003 and 2004, we use the SCIAMACHY spatial fields to gap fill the record. **Figure 5** shows that
SCIAMACHY had some substantially large biases compared to the ozonesondes in 2006, 2007 and 2008 but
was reasonable for other years. Therefore, we use the global distributions from SCIAMCHY for both years
but scale them to expected values between 2002 and 2005. This is achieved by getting the globally weighted
(based on surface area) $LTCO_3$ average for GOME-1 (2002 with GOME-1 with the OMI offset applied) and
OMI (2005) and the SCIAMACHY (2003-2004). Based on the difference between 2002 and 2005, an annual
linear global scaling is applied in 2003 and 2004 for the SCIAMACHY spatial fields. Thus, we have developed a
harmonised $LCTO_3$ record between 1996 and 2017. Examples of the harmonised data for Europe and East
Asia are shown in **Figure 6**. Overall, there is non-linear variability in the two regional time-series where red
and blue show the GOME-1 and OMI $LTCO_3$ time series and then black shows where they have been merged
(including SCIAMACHY for 2003 and 2004). For Europe (East Asia), the seasonal cycle ranges between 10.0
(13.0) and 30.0 (27.0) DU, respectively, with annual average values between 18.0 (18.0) and 22.0 (21.0) DU.

### 3.5. Lower Tropospheric Column Ozone Temporal Variability
The harmonised RAL Space data set can now be used to investigate decadal scale spatiotemporal variability
in $LTCO_3$. **Figure 7** shows the global long-term (1996-2017) average in $LTCO_3$ and the 5-year average
anomalies for 1996-2000, 2005-2009 and 2013-2017. In the long-term average (**Figure 7a**), there is clear SH
to NH $LTCO_3$ gradient with background values of 13.0-17.0 DU and 20-23.0 DU, respectively. There are
hotspots over East Asia, the Middle East/Mediterranean and northern India of 24.0-25.0 DU. The largest SH
$LTCO_3$ values (20.0-22.0 DU) are between 30-15°S spanning southern Africa, the Indian Ocean and Australia.
Minimum $LTCO_3$ values (<12.0 DU) are over the Himalayas (due to topography) and the tropical oceans. As
shown in **Figure 2**, there is sufficient information (e.g. $LTCO_3$ DOFS mostly > 0.5) in the tropics and mid-
latitudes for the instruments used to form the merged $LTCO_3$ data. This provides confidence in this merged
$LTCO_3$ record for long-term temporal analysis. Note, the SAA has been masked out in all the panels. The
1996-2000 anomaly map (**Figure 7b**) shows values to be similar (i.e. -1.0 to 1.0 DU) with respect to the 1996-
2017 mean between 30°N and 60°N. A similar relationship occurs at approximately 30°S. However, in tropics
and NH sub-tropics (15°S to 30°N), the anomalies are more negative, ranging between approximately -3.0
and -1.0 DU. The green polygon-outlined regions show where the 1996-2000 $LTCO_3$ average represents a
substantial difference (p-value < 0.05) from the long-term average. This is based on the Wilcoxon rank test
(WRT), which is the nonparametric counterpart of the Student t-test that relaxes the constraint on normality
of the underlying distributions (Pirovanoet al., 2012). As well as this tropical band, the 60-45°S band shows
anomalies of a similar magnitude. In the 2005-2009 anomaly map (**Figure 7c**), there are widespread, though
non-substantial, anomalies of -1.5.0 to 0.0 DU. There are small clusters of substantial anomalies (e.g.
southern Africa at -2.0 to -1.0 DU and over the Bering Sea between 1.0 and 2.0 DU) but with limited spatial
coherence. In the 2013-2017 anomaly map (**Figure 7d**), they remain small $LTCO_3$ anomalies in the northern
mid-latitudes (-1.0 to 1.0 DU). A similar pattern occurs in the southern sub-tropics and mid-latitudes, though
the anomalies are larger peaking at 1.5 DU around 60-45°S (some have p-values < 0.05). However, in the
tropics and sub-tropics (15°S-30°N), there are positive anomalies of 1.0 to 2.0 DU throughout the region,
peaking at 2.0-2.5 DU over Africa.
Overall, these anomalies suggest there has been limited change in $LTCO_3$, between 1996 and 2017, in the NH
mid-latitudes (e.g. as can be seen for Europe and East Asia in **Figure 6**). Unfortunately, the SAA masks any
useful information on LTCO$_3$ over South America, but generally there has been a moderate LTCO$_3$ increase in
the SH mid-latitudes. The largest and most substantial changes have been in the tropics and sub-tropics (i.e.
15°S to 30°N) switching from sizeable negative anomalies (-2.0 to -1.0 DU) in the 1996-2000 LTCO$_3$ average
to positive anomalies (1.0-2.0 DU) in the 2013-2017 LTCO$_3$. **Figure 8** shows the difference between the 2013-
2017 and 1996-2000 averages. Over the tropics/sub-tropics (15°S-30°N), the largest increases (p-values <
0.05) of 3.0 to 5.0 DU occur peaking Africa, India and South-East Asian (>5.0 DU). Thus, showing a large-scale
increase in tropical LTCO$_3$ between 1996 and 2017. In the NH mid-latitudes, the absolute LTCO$_3$ differences
are relatively small (-1.0 to -1.5 DU) but there are consistent, though some negative differences (generally -
2.0 and -1.0 DU) are over North America and Russia. In the SH mid-latitudes, there has been moderate
increase in LTCO$_3$ of 2.0-3.5 DU. However, southern Africa shows more localised decreases of up to 3.0 DU
and non-substantial differences at 30°S across the Indian Ocean. The ozonesondes are consistent with
satellite 1996-2000 and 2013-2017 average LTCO$_3$ differences. In the tropics, the majority of ozonesonde
sites show increases between these two periods ranging between 0.5 and 5.0 DU. Over Europe (i.e. northern
mid-latitudes), the ozonesonde LTCO$_3$ differences range between -0.5 and 0.5 DU suggesting limited LTCO$_3$
change over time.
### 3.6.  Long-term LTCO$_3$ Trends
In line with TOAR-II, we have added additional metrics on the temporal change in LTCO$_3$ over the merged
instrument record. Here, we have calculated the linear trends in LTCO$_3$ in 15° latitude bins between 60°S-
60°N along with the 95% confident range and associated p-values (see **Table 2**). In the tropical latitudes
(15°S-30°N), all the linear trends show substantial increasing trends (2.89-4.12 DU/decade) between 1996
and 2017; all with p-values tending to 0.0. This is consistent with the LTCO$_3$ positive differences (3.0-5.0 DU)
between the 1996-2000 and 2013-2017 averages (**Figure 8**). In the northern mid-latitudes (30-60°N), there
are smaller positive trends (1.33 and 0.49 DU/decade) but the 95% confidence values intersect with 0.0 and
have larger p-values. Again, this is consistent with the near-zero differences between the 1996-2000 and
2013-2017 averages (**Figure 8**). In the southern mid-latitudes (30-60°S), the trends are substantially positive
(1.85 and 4.49 DU/decade) with near-zero p-values. Again, this is consistent with the substantial differences
(2.0-4.0 DU) between the 1996-2000 and 2013-2017 averages. The 15-30°S trend is small at 0.94 DU/decade
with a moderate p-value of 0.35, indicating this not to be a substantial trend.
### 4.  Discussion and Conclusions
Multiple studies have used satellite records to investigate change in TCO$_3$ in recent decades. Gaudel et al.,
(2018) used a range of UV-Vis and IR TCO$_3$ products between 2005 and 2016. The UV-Vis sounders generally
show substantial positive trends (0.1-0.8 DU/yr) in the tropics/sub-tropics and a mixed response in the mid-
latitudes. The IR instruments typically showed significant decreasing trends (-0.5 to -0.2 DU/yr) in
background regions and isolated regions of substantial TCO$_3$ enhancements. Ziemke et al., (2019) used a
long-term merged record of TCO$_3$ from the Total Ozone Mapping Spectrometer (TOMS) and Ozone
Monitoring Instrument/Microwave Limb Sounder (OMI-MLS) between 1979 and 2016. Over this period, they
found significant increases of TCO$_3$ of 1.5 to 6.5 DU, especially over India and East Asia. Heue et al., (2016)
used a long-term tropical TCO$_3$ record (GOME, SCIAMACHY, OMI, GOME-2A and GOME-2B) finding
significant increases (0.5-2.0 DU/decade) over central Africa and the South Atlantic. However, the study by
Wespes et al., (2018) from IASI (an IR sounder) indicated that TCO$_3$ decreased between 2008 and 2017 by -
0.5 to -0.1 DU/yr. Gaudel et al., (2018) reported similar TCO$_3$ tendencies using two IASI products (IASI-FORLI
and IASI-SOFRID). However, Boynard et al., (2018) and Wespes et al., (2018) report a step-change in 2010 in
the IASI-FORLI O$_3$ data which could influence observed long-term trends. Therefore, studies using IR
products available to TOAR-I and Wespes (2018) are no longer considered reliable.
In this study, for the first time we analysed long-term changes in LTCO$_3$ using a merged satellite UV-Vis
sounder record. Overall, we found that LTCO$_3$ was lower (by 1.0-3.0) in the tropics between 1996 and 2000
in comparison to the long-term average (i.e. 1996-2017). Similar LTCO$_3$ values exist between the 2005-2009
and long-term averages, while the 2013-2017 average shows substantially larger tropical values (1.0-2.5 DU)
than the long-term average. Therefore, this tropical increase (3.0-5.0 DU) in LTCO$_3$ between 1996 and 2017 is
consistent with other reported increases in TCO$_3$. A similar consistency is found in the NH mid-latitudes, with
minimal changes in LTCO$_3$ observed here and in trends in TCO$_3$ reported in Gaudel et al., (2018) and Ziemke
et al., (2019).  Sizable LTCO$_3$ increases in the SH mid-latitudes are also consistent with Gaudel et al., (2018)
and Ziemke et al., (2019), though they differ from IASI retrieved TCO$_3$ trends as reported by Wespes et al
(2018). Overall, the long-term changes in LTCO$_3$ reported here and the literature TCO$_3$ trends from satellite
UV products are comparable in regard to latitude dependence and direction. It therefore seems that the
positive tendencies in TCO$_3$ reported in the literature from UV soundings over the satellite-era are associated
with, and could be driven by, changes occurring in LTCO$_3$.
For future work, a detailed study is required to disentangle the reported TCO$_3$ and LTCO$_3$ trends reported by
UV-Vis and IR sounders, which would benefit from satellite level-2 data produced from level-1 data sets
which are more uniform over time along with other improvements. This can potentially be done also by
using a 3D atmospheric chemistry model (ACM) to investigate the changes in lower and upper tropospheric
ozone, and application of the satellite AKs (i.e. the vertical sensitivity of the different satellite products) to
the model from the different sounders to establish how satellite vertical sensitivity potentially changes the
simulated TO$_3$ tendency of the model. An ACM would also be a useful tool to help diagnose the importance
of LTCO$_3$ contributions to the TCO$_3$ tendencies, and which processes might be driving any spatiotemporal
changes (e.g. surface emissions, atmospheric chemistry/surface deposition, stratospheric-tropospheric O$_3$
exchanges etc.). Finally, together with improved, extended reprocessed versions of the data sets used in this
study, the launch of the Sentinel 5 – Precursor (S5P) satellite (in October 2017) can be used to extend the
merged data record of LTCO$_3$, along with new polar orbiting platforms such as Sentinel-5 and IASI-NG
instruments on future EUMETSAT MetOp-Second Generation satellites.
**Acknowledgements:**
This work was funded by the UK Natural Environment Research Council (NERC) by providing funding for the
National Centre for Earth Observation (NCEO, award reference NE/R016518/1) and funding from the
European Space Agency (ESA) Climate Change Initiative (CCI) post-doctoral fellowship scheme (contract
number 4000137140). Anna Maria Trofaier (ESA Climate Office) provided support and advice throughout the
fellowship.
**Conflicting Interests**
The authors declare that they have no conflicts of interest.
**Data Availability**
The RAL Space satellite data is available via the NERC Centre for Environmental Data Analysis (CEDA) Jasmin
platform subject to data requests. The RAL Space satellite data will be uploaded to the Zenodo open access
portal (https://zenodo.org/) if this manuscript is accepted for publication in ACP after the peer-review
process. The ozonesonde data for WOUDC, SHADOZ and NOAA is available from https://woudc.org/,
https://tropo.gsfc.nasa.gov/shadoz/ and https://gml.noaa.gov/ozwv/ozsondes/.
**Author Contributions**
RJP, MPC and BJK conceptualised and planned the research study. RJP and MAP analysed the satellite data
provided by RAL Space (BJK, RS, BGL) with support from BJK, RS and BGL. MPC, SD and CR provided scientific
advice, while WF and RR provided technical support. RJP prepared the manuscript with contributions from
all co-authors.

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

| Data Provider | Satellite Profile Products & Version | Product Link | Data Range | Data Size |
|---|---|---|---|---|
| RAL Space | OMI–fv214 | http://www.ceda.ac.uk/ | 2004-2018 | 1442 GB |
| RAL Space | GOME-2A-fv300 | http://www.ceda.ac.uk/ | 2007-2019 | 1007 GB |
| RAL Space | GOME-1-fv301 | http://www.ceda.ac.uk/ | 1995-2011 | 703 GB |
| RAL Space | SCIAMACHY-fv300 | http://www.ceda.ac.uk/ | 2002-2012 | 718 GB |

**Table 1:** *List of RAL Space level-2 satellite ozone profile data sets.*

| Latitude Band | LTCO$_3$ Trend (DU/decade) (95% Confidence Interval) | LTCO$_3$ Trend (ppbv/decade) (95% Confidence Interval) | p-values |
|---|---|---|---|
| 60°S ≤ Latitude < 45°S | 4.49 (2.51, 6.48) | 10.37 (5.79, 14.95) | 0.00 |
| 45°S ≤ Latitude < 30°S | 1.85 (0.11, 3.59) | 4.27 (0.26, 8.28) | 0.03 |
| 30°S ≤ Latitude < 15°S | 0.94 (-1.05, 2.93) | 2.17 (-2.42, 6.76) | 0.35 |
| 15°S ≤ Latitude < 0° | 2.89 (1.27, 4.52) | 6.68 (2.94, 10.43) | 0.00 |
| 0° ≤ Latitude < 15°N | 3.93 (3.13, 4.72) | 9.06 (7.23, 10.89) | 0.00 |
| 15°N ≤ Latitude < 30°N | 4.12 (3.25, 4.97) | 9.50 (7.51, 11.48) | 0.00 |
| 30°N ≤ Latitude < 45°N | 1.33 (-0.34, 3.01) | 3.08 (-0.78, 6.95) | 0.11 |
| 45°N ≤ Latitude < 60°N | 0.49 (-1.14, 2.13) | 1.14 (-2.64, 4.91) | 0.55 |

**Table 2**: *LTCO$_3$ trends (DU/decade and ppbv/decade) for latitude bands (15° bins) between 60°S and 60°N. The 95% confidence intervals of the trends are shown in brackets. The trend p-values are also shown.*

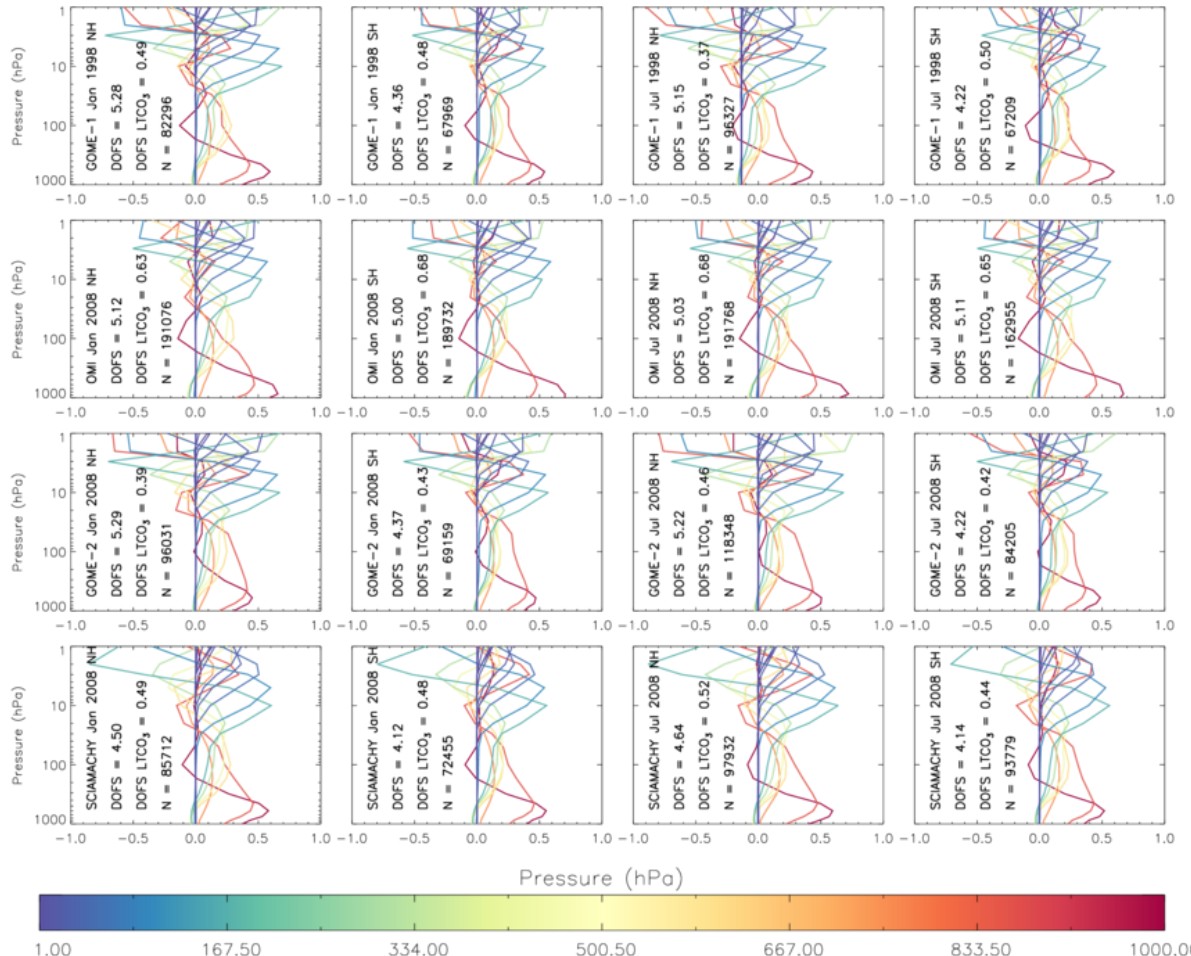

531

**Figure 1**: *Average averaging kernels (AKs) for the instruments listed in **Table 1** for the northern and southern hemispheres (60°S-60°N) in January and July of 2008 (1998 for GOME-1). The average degrees of freedom of signal (DOFS) is shown as is DOFS LTCO$_3$ which represents the DOFS in the lower tropospheric column ozone (LTCO$_3$). N represents the number of retrievals in each average AK average.*

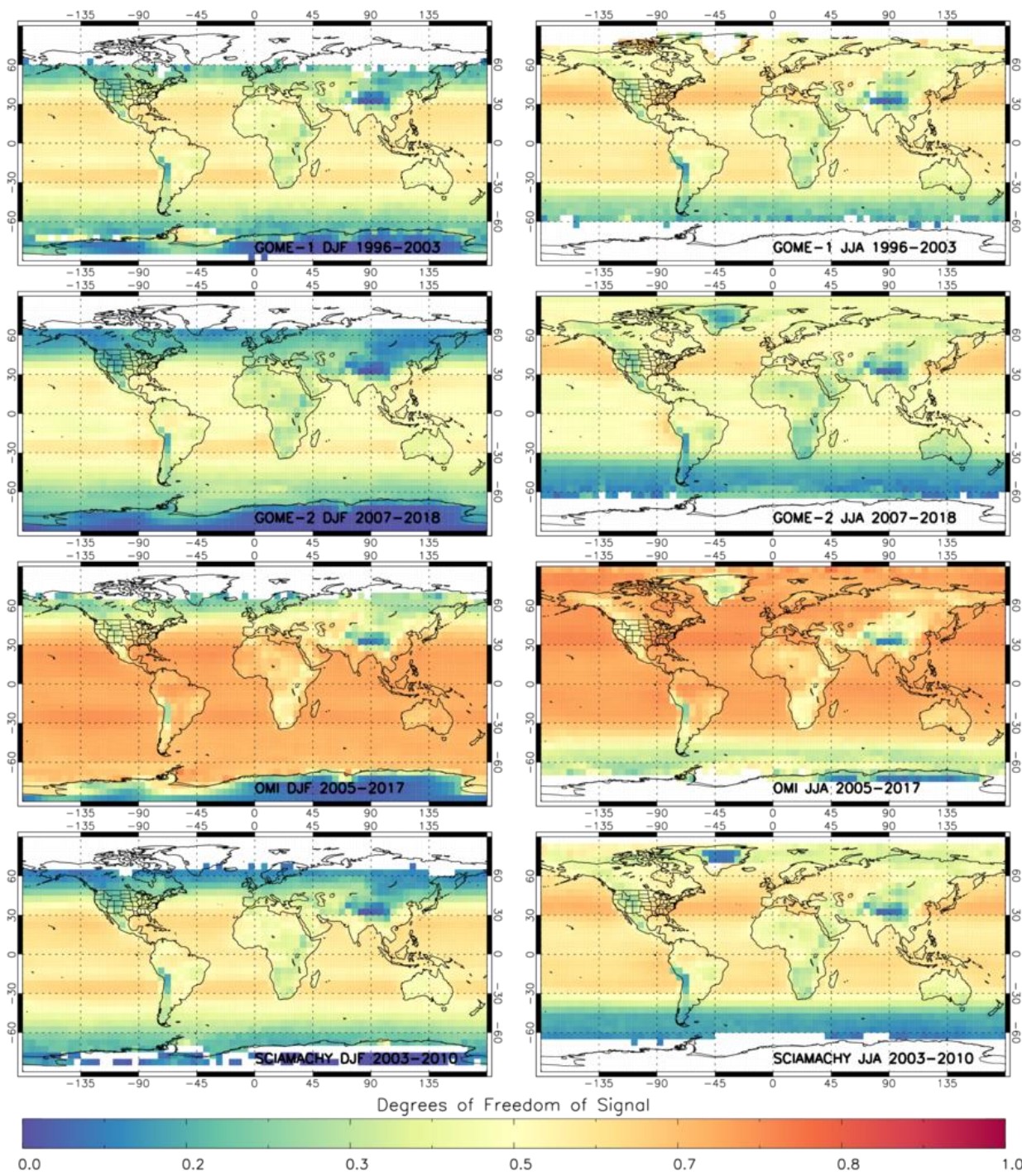

536
**Figure 2**: *Seasonal distributions of LTCO₃ degrees of freedom of signal (DOFS) in DJF and JJA for GOME-1,*
*GOME-2, OMI and SCIAMACHY averaged over the full record for each instrument.*

539

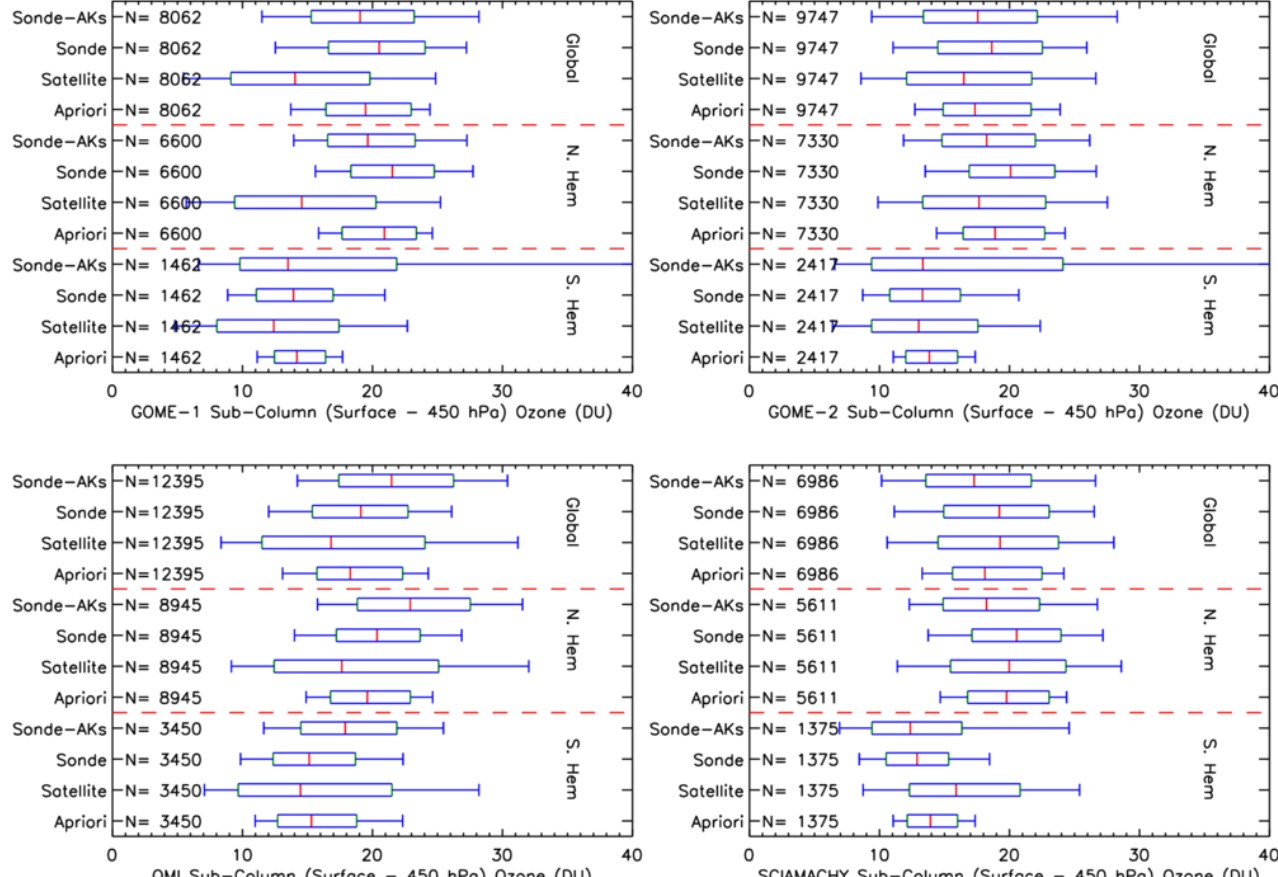

**Figure 3:** Box and whisker distributions of $LTCO_3$ from satellite, apriori, ozonesonde (Sonde) and ozonesonde with AKs applied (Sonde-AKs) for co-located samples (i.e. satellite and ozonesonde profiles co-located within 6-hours and 500 km). This is done for GOME-1 (top-left), GOME-2 (top-right), OMI (bottom-left) and SCIAMACHY (bottom-right) on a global, southern hemispheric and northern hemispheric basis over their respectively records. Red dashed lines separate the box and whisker distributions for each region. The red, green and blue vertical lines represent the $50^{th}$, $25^{th}$ & $75^{th}$ and $10^{th}$ & $90^{th}$ percentiles. N represents the sample size.

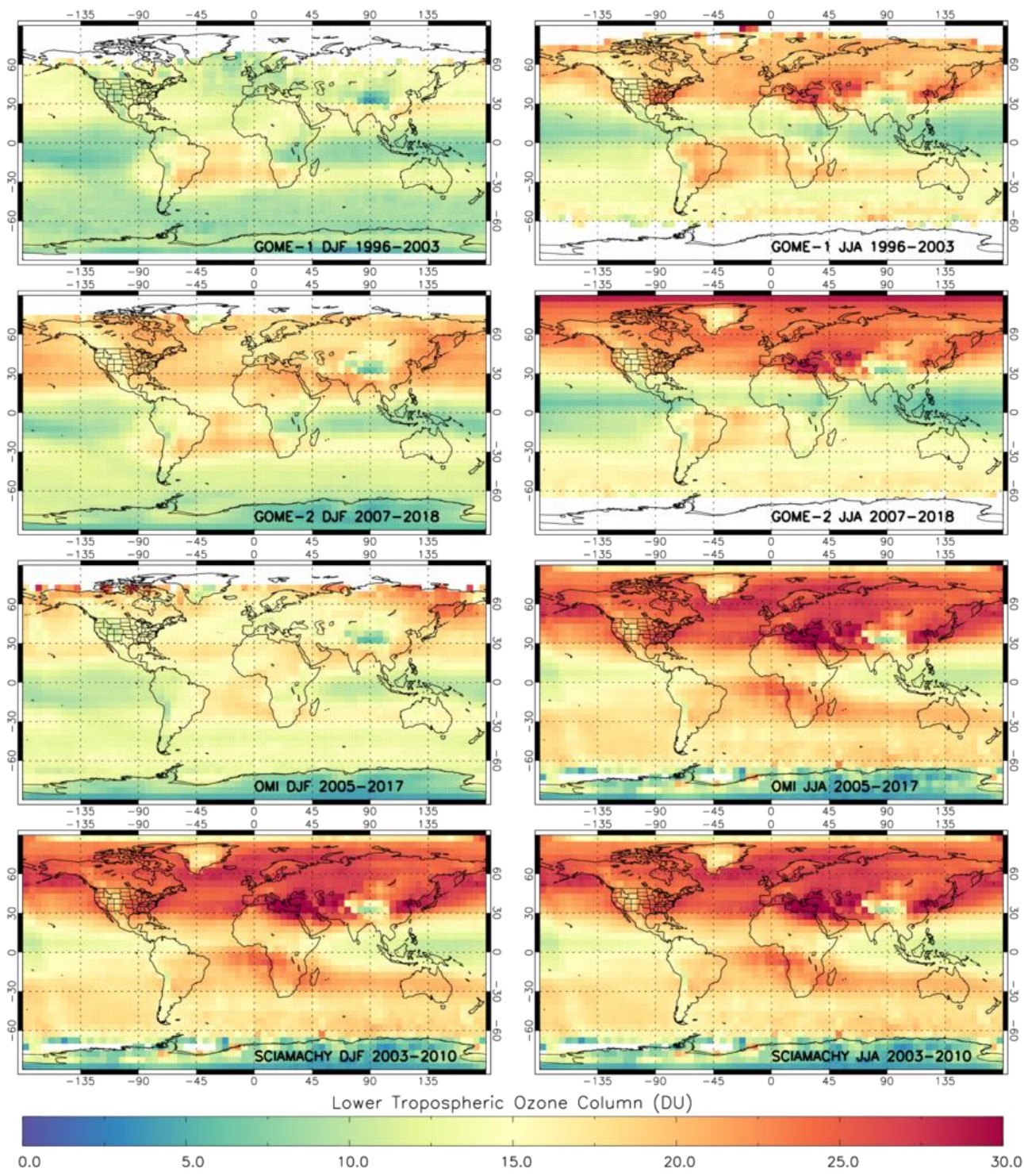

**Figure 4**: Seasonal distributions of LTCO$_3$ in December-January-February (DJF) and June-July-August (JJA) for OMI, GOME-1, GOME-2 and SCIAMACHY averaged over the full record for each instrument.

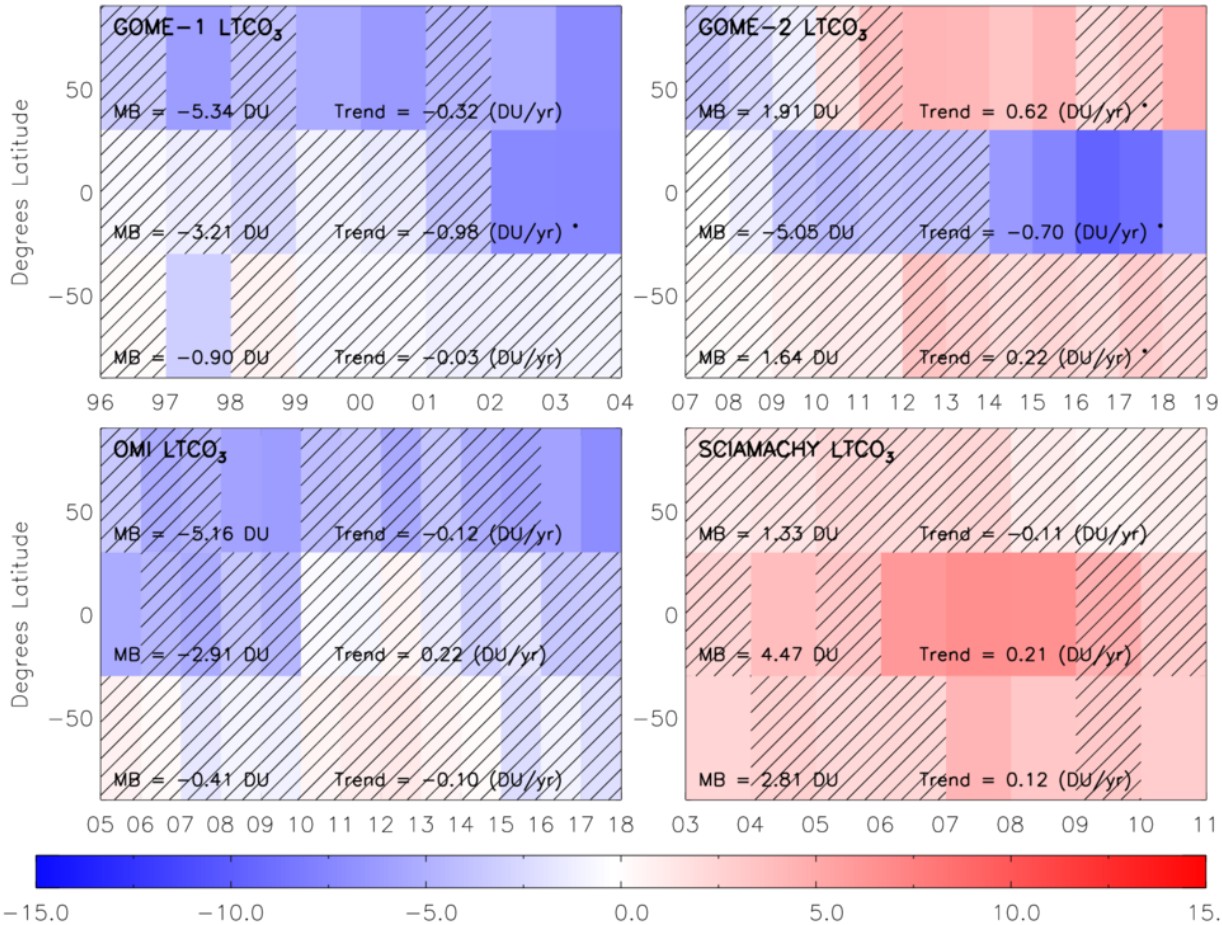

**Figure 5**: *Latitudinal-annually varying satellite-sonde, with AKs applied, LTCO$_3$ (DU) median (50$^{th}$ percentile) biases. Hatched regions show where the spread in the 25$^{th}$ and 75$^{th}$ percentiles intersect with 0.0. The mean bias (MB) and trend are for the full time series of each hemisphere. The \* for the trend term indicates that it has a p-value < 0. 05. The latitude bands are 90-30°S, 30°S-30°N and 30-90°N.*

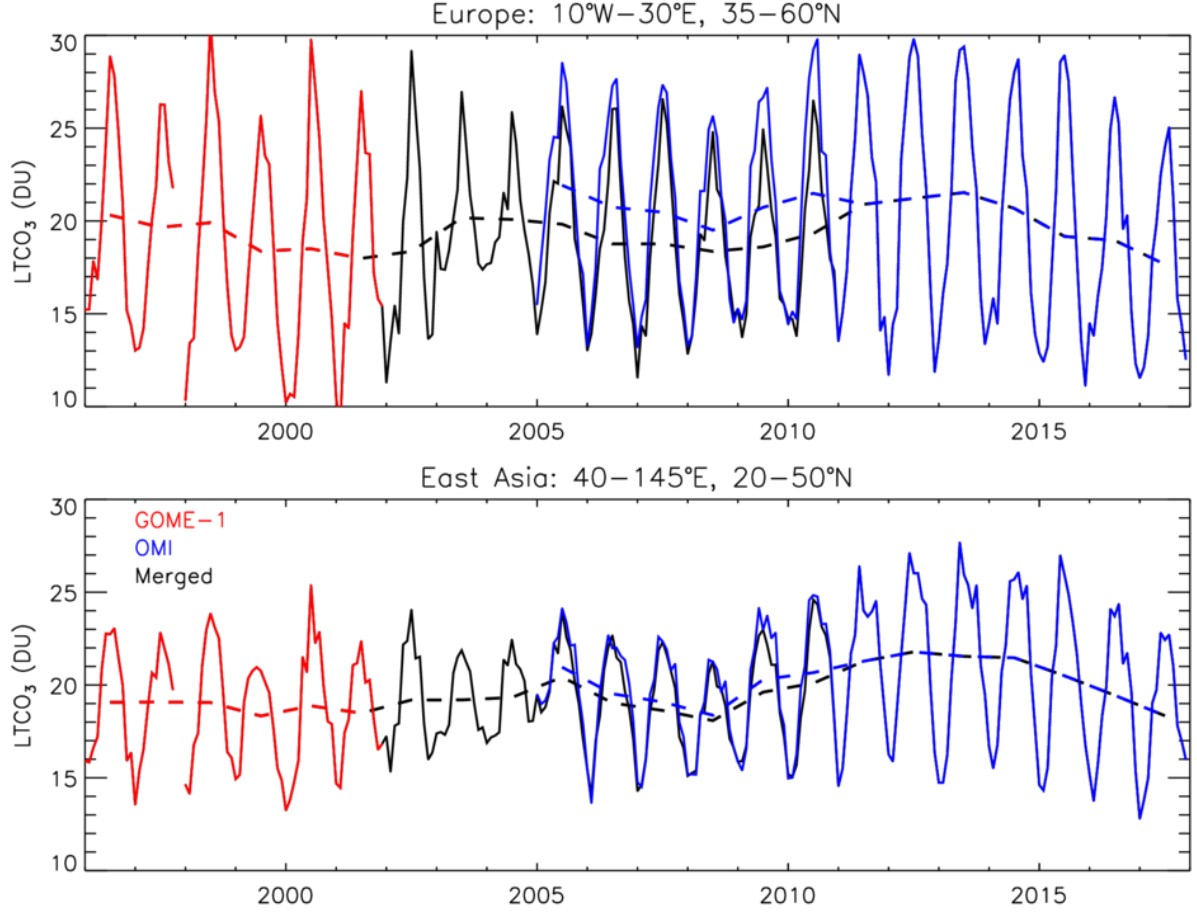


**Figure 6**: *Examples of the merged LTCO₃ (DU) data set for Europe and East Asia. The GOME-1, OMI and*
*merged time series are shown in red, blue and black, respectively. The merged record also includes globally*
*scaled LTCO₃ data from SCIAMACHY for 2003 and 2004. Dashed lines represent the annual averages and the*
*monthly mean time-series are solid lines.*

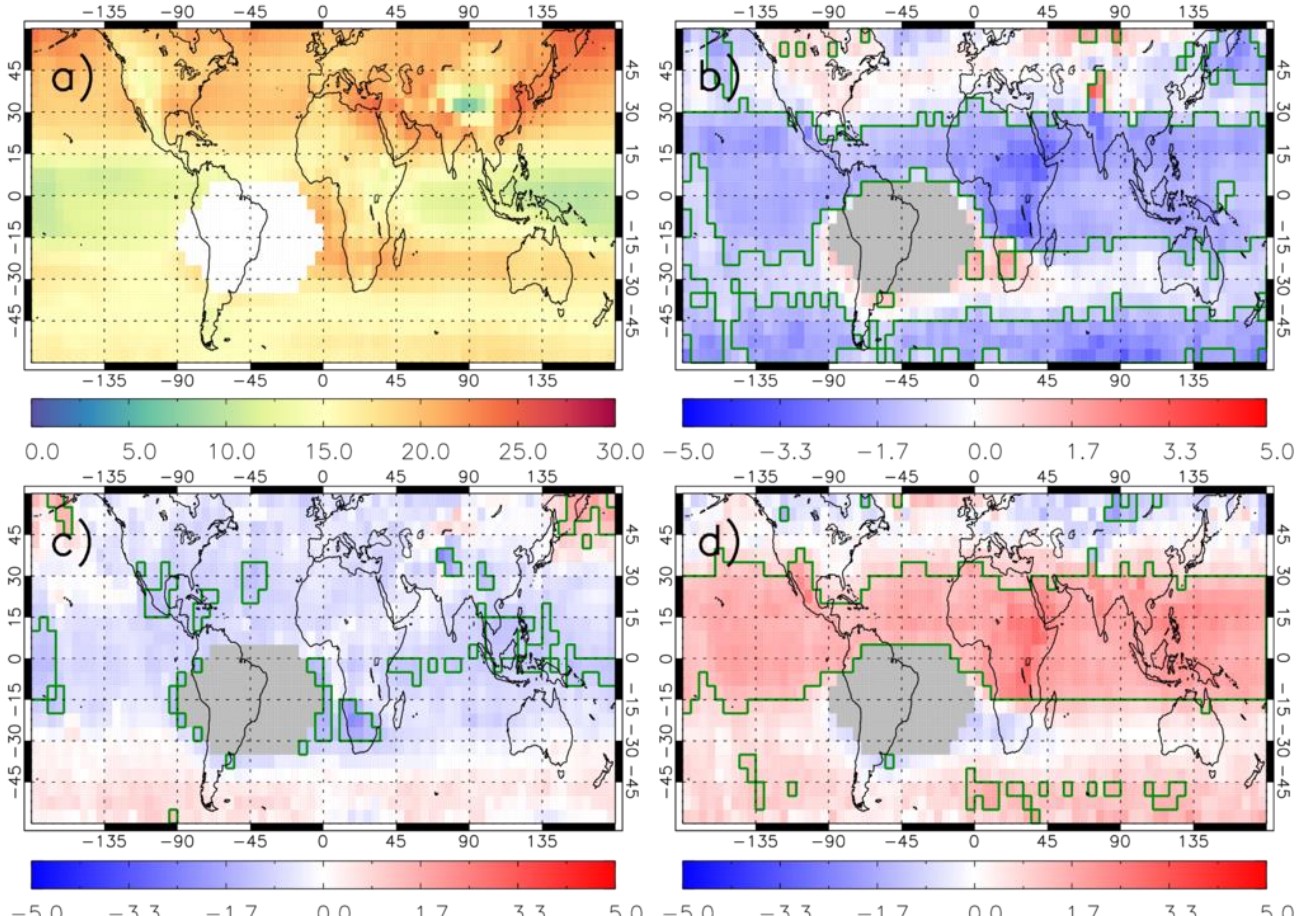

**Figure 7**: LTCO$_3$ (DU) merged data set from GOME-1 (1996-2002), SCIAMACHY (2003-2004) and OMI (2005-2017). a) 1996-2017 long-term average, b) 1996-2000 average anomaly, c) 2005-2009 average anomaly and d) 2013-2017 average anomaly. Anomalies are relative to the long-term average (panel a). Green polygon-outlined regions show significant anomalies (95% confidence level and where the absolute anomaly > 1.0 DU) from the long-term average using the Wilcoxon Rank Test. White/grey pixels are where the South Atlantic Anomaly influence on retrieved LTCO$_3$ has been masked out.

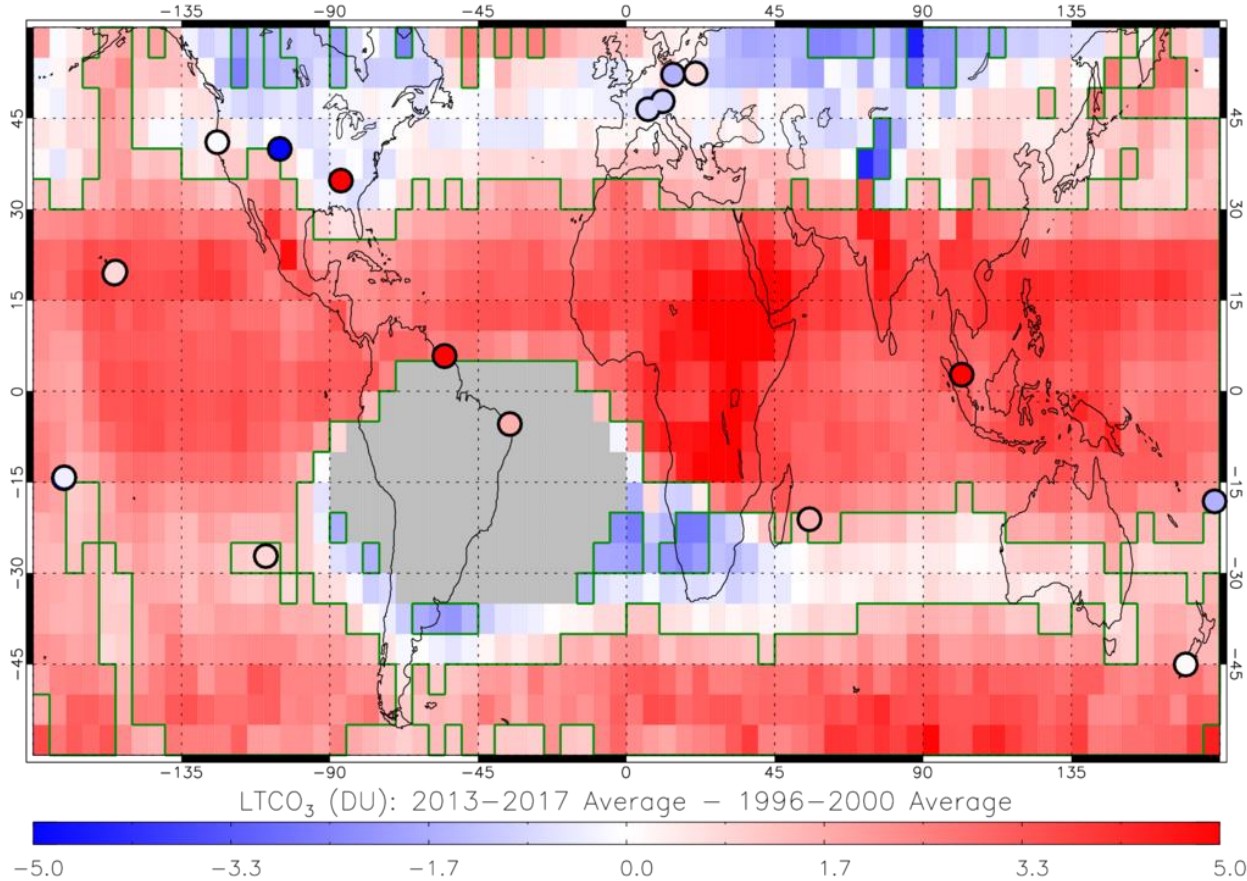


***Figure 8****: LTCO3 (DU) merged data set from GOME-1 (1996-2002), SCIAMACHY (2003-2004) and OMI (2005-*
*2017) where the difference between the 2013-2017 average and 1996-2000 average is shown. Green*
*polygon-outlined regions show substantial differences (95% confidence level and where the absolute*
*difference > 1.0 DU) using the Wilcoxon Rank Test. Grey pixels are where the South Atlantic Anomaly*
*influence on retrieved LTCO$_3$ has been masked out. Circles show differences in ozonesonde LTCO$_3$ (DU) over*
*the same time periods as the merged satellite record.*