# Peer review of "Investigation of spatial and temporal variability in lower tropospheric ozone from RAL Space UV-Vis satellite products"

_EGUsphere, 2023_

## Community Comment (CC1)

Comments by Owen R. Cooper (TOAR Scientific Coordinator of the Community Special Issue) on:

**Investigation of spatial and temporal variability in lower tropospheric ozone from RAL Space UV-Vis satellite products**

Richard J. Pope (corresponding author), Brian J. Kerridge, Richard Siddans, Barry G. Latter, Martyn P. Chipperfield, Wuhu Feng, Matilda A. Pimlott, Sandip S. Dhomse, Christian Retscher, and Richard Rigby

This manuscript was submitted to ACP as part of the TOAR-II Community Special Issue
https://doi.org/10.5194/egusphere-2023-1172, 2023

This review is by Owen Cooper, TOAR Scientific Coordinator of the TOAR-II Community Special Issue. I, or a member of the TOAR-II Steering Committee, will post comments on all papers submitted to the TOAR-II Community Special Issue, which is an inter-journal special issue accommodating submissions to six Copernicus journals:  ACP (lead journal), AMT, GMD, ESSD, ASCMO and BG. The primary purpose of these reviews is to identify any discrepancies across the TOAR-II submissions, and to allow the author teams time to address the discrepancies.  Additional comments may be included with the reviews.

**General Comments:**

TOAR-II has produced two guidance documents to help authors develop their manuscripts so that results can be consistently compared across the wide range of studies that will be written for the TOAR-II Community Special Issue.  Both guidance documents can be found on the TOAR-II webpage: https://igacproject.org/activities/TOAR/TOAR-II

*The TOAR-II Community Special Issue Guidelines*:   In the spirit of collaboration and to allow TOAR-II findings to be directly comparable across publications, the TOAR-II Steering Committee has issued this set of guidelines regarding style, units, plotting scales, regional and tropospheric column comparisons, tropopause definitions and best statistical practices.

*The TOAR-II Recommendations for Statistical Analyses*:  The aim of this guidance note is to provide recommendations on best statistical practices and to ensure consistent communication of statistical analysis and associated uncertainty across TOAR publications. The scope includes approaches for reporting trends, a discussion of strengths and weaknesses of commonly used techniques, and calibrated language for the communication of uncertainty.

Discussion of trends:
The expression "statistically significant" is used throughout the submitted manuscript, however this expression is now recognized as being problematic and it should be abandoned and replaced by the more useful method of reporting all trends (with uncertainty, e.g. 95% confidence intervals) and all $p$-values, followed by a discussion of the trends and the author's opinion regarding their confidence in the trend values.  This advice comes from a highly influential paper by Wasserstein et al. (2019), published in the journal, The American Statistician, that has already been cited over 1400 times (according to Web of Science).  This advice was adopted by the first phase of TOAR (Tarasick et al., 2019) and will also be used

by TOAR-II. Some other recent papers on ozone trends that have taken this advice are: Chang et al., 2020; Cooper et al., 2020; Gaudel et al., 2020; Chang et al., 2022; Wang et al., 2022. Because these papers report all trend values, uncertainties, and all p-values, and also discuss the trend results, there is no confusion regarding the findings, and one does not even notice that the term "statistically significant" is not used at all. Table 3 of the TOAR-II statistical guidelines provides calibrated language for describing trends and uncertainty, similar to the approach of IPCC.

Below is a figure from Chapter 2 of IPCC AR6 WG-I (Gulev et al., 2021) summarizing observed global ozone trends. TOAR-II will produce a similar figure from all recent ozone trend studies published in the TOAR-II Community Special Issue (as well as from studies not in the special issue). Trends from all new satellite studies can be added to the right-hand panel, but the trends must be reported in units of ppbv/decade, with p-values and 95% confidence intervals. Your study currently reports ozone changes in Dobson units from one 5-year period to the next. Could you report these values as trends in units of ppbv/decade? You can choose whichever latitude bands you like when reporting the zonal trends, but useful intervals would be 10 or 15 degrees.

[Figure]

**Figure 2.8 | Surface and tropospheric ozone trends. (a)** Decadal ozone trends by latitude at 28 remote surface sites and in the lower free troposphere (650 hPa, about 3.5 km) as measured by IAGOS aircraft above 11 regions. All trends are estimated for the time series up to the most recently available year, but begin in 1995 or 1994. Colours indicate significance (p-value) as denoted in the in-line key. See Figure 6.5 for a depiction of these trends globally. **(b)** Trends of ozone since 1994 as measured by IAGOS aircraft in 11 regions in the mid-troposphere (700–300 hPa; about 3–9 km) and upper troposphere (about 10–12 km), as measured by IAGOS aircraft and ozonesondes. **(c)** Trends of average tropospheric column ozone mixing ratios from the TOST composite ozonesonde product and three composite satellite products based on TOMS, OMI/MLS (Sat1), GOME, SCIAMACHY, OMI, GOME-2A, GOME-2B (Sat2), and GOME, SCIAMACHY, GOME-II (Sat3). Vertical bars indicate the latitude range of each product, while horizontal lines indicate the *very likely* uncertainty range. Further details on data sources and processing are available in the chapter data table (Table 2.SM.1).

This paper reports satellite ozone retrievals for the lower troposphere, which is defined as the surface to 450 hPa.  Typically, when one thinks of the lower troposphere, the layer from the surface to about 700 or 600 hPa comes to mind, for example the IASI-GOME2 product focuses on the lower troposphere and spans the layer from the surface to 3 km. The layer from the surface to 450 hPa also includes the region of the atmosphere that is typically thought of as the mid-troposphere (600-400 hPa).  In cold months at high latitudes the tropopause is often as low as 300 hPa, in which case the surface-450 hPa layer spans most of the troposphere.  To avoid confusion with other products/definitions, and to highlight that the new RAL product also spans the mid-troposphere, can the product be referred to as lower-mid tropospheric?

Line 321
The paper sates that IR satellite instruments tend to report negative ozone trends globally, but I don't think that such a generalization can be made.  Gaudel et al (2018) compared two IASI ozone products and the trends don't always agree.  As shown in Figure 23, In the northern tropics IASI-FORLI shows a weak positive trend (IASI-SOFRID is negative), while at northern mid-latitudes IASI-SOFRID shows no trend but IASI-FORLI is strongly negative.  After the publication of Gaudel at al. (2018), Boynard et al. (2018) reported a negative drift in the IASI-A instrument. If this drift is taken into account in updated IASI trends then the negative trends may not be so strong.

Ziemke et al. (2019) found a positive drift in the OMI/MLS product and added a correction to their final product to account for the drift.  The submitted paper reports drift values for all products but only found the GOME-II drift to be of concern and rejected that particular dataset.  The drift for OMI was reported as 0.22 DU/yr in the tropics, but deemed insignificant. But still, after 15 years the drift would add up to 3.3 DU, which is more than 10-20% of the tropical tropospheric ozone column. Was a correction applied to account for drift?

**Minor Comments:**

Abstract, Line 35
"While GOME-2…" would sound better as "However, GOME-2…"

Line 44
Here and throughout, references to IPCC AR5 should be updated with References to IPCC AR6. For this work, the relevant chapters of IPCC AR6 WG-I are Chapters 2 (Gulev et al., 2021), 6 (Szopa et al., 2021) and 7 (Forster et al., 2021). For example, the latest IPCC estimate of ozone's effective radiative forcing (ERF) is +0.47 [0.24 to 0.70] W m$^{-2}$ (1750-2019, tropospheric + stratospheric ozone) (Forster et al., 2021).

**References**
 Boynard, A., Hurtmans, D., Garane, K., Goutail, F., Hadji-Lazaro, J., Koukouli, M. E., Wespes, C., Vigouroux, C., Keppens, A., Pommereau, J.-P., Pazmino, A., Balis, D., Loyola, D., Valks, P., Sussmann, R., Smale, D., Coheur, P.-F., and Clerbaux, C.: Validation of the IASI FORLI/EUMETSAT ozone products using satellite (GOME-2), ground-based (Brewer–Dobson, SAOZ, FTIR) and ozonesonde measurements, Atmos. Meas. Tech., 11, 5125–5152, https://doi.org/10.5194/amt-11-5125-2018, 2018.

Chang, K.-L., et al. (2020), Statistical regularization for trend detection: An integrated approach for detecting long-term trends from sparse tropospheric ozone profiles, Atmos. Chem. Phys., 20, 9915–9938, https://doi.org/10.5194/acp-20-9915-2020

Chang, K.-L., O. R. Cooper, A. Gaudel, M. Allaart, G. Ancellet, H. Clark, S. Godin-Beekmann, T. Leblanc, R. Van Malderen, P. Nédélec, I. Petropavlovskikh, W. Steinbrecht, R. Stübi, D. W. Tarasick, C. Torres (2022), Impact of the COVID-19 economic downturn on tropospheric ozone trends: an uncertainty weighted data synthesis for quantifying regional anomalies above western North America and Europe, *AGU Advances, 3*, e2021AV000542. https://doi.org/10.1029/2021AV000542

Cooper, et al. 2020. Multi-decadal surface ozone trends at globally distributed remote locations. Elem Sci Anth, 8: 23. DOI: https://doi.org/10.1525/elementa.420

Forster, P., T. Storelvmo, K. Armour, W. Collins, J.-L. Dufresne, D. Frame, D.J. Lunt, T. Mauritsen, M.D. Palmer, M. Watanabe, M. Wild, and H. Zhang, 2021: The Earth's Energy Budget, Climate Feedbacks, and Climate Sensitivity. In Climate Change 2021: The Physical Science Basis. Contribution of Working Group I to the Sixth Assessment Report of the Intergovernmental Panel on Climate Change [Masson-Delmotte, V., P. Zhai, A. Pirani, S.L. Connors, C. Péan, S. Berger, N. Caud, Y. Chen, L. Goldfarb, M.I. Gomis, M. Huang, K. Leitzell, E. Lonnoy, J.B.R. Matthews, T.K. Maycock, T. Waterfield, O. Yelekçi, R. Yu, and B. Zhou (eds.)]. Cambridge University Press, Cambridge, United Kingdom and New York, NY, USA, pp. 923–1054, doi:10.1017/9781009157896.009

Gaudel, A., et al. (2020), Aircraft observations since the 1990s reveal increases of tropospheric ozone at multiple locations across the Northern Hemisphere. Sci. Adv. 6, eaba8272, DOI: 10.1126/sciadv.aba8272

Gulev, S.K., P.W. Thorne, J. Ahn, F.J. Dentener, C.M. Domingues, S. Gerland, D. Gong, D.S. Kaufman, H.C. Nnamchi, J. Quaas, J.A. Rivera, S. Sathyendranath, S.L. Smith, B. Trewin, K. von Schuckmann, and R.S. Vose, 2021: Changing State of the Climate System. In Climate Change 2021: The Physical Science Basis. Contribution of Working Group I to the Sixth Assessment Report of the Intergovernmental Panel on Climate Change [Masson-Delmotte, V., P. Zhai, A. Pirani, S.L. Connors, C. Péan, S. Berger, N. Caud, Y. Chen, L. Goldfarb, M.I. Gomis, M. Huang, K. Leitzell, E. Lonnoy, J.B.R. Matthews, T.K. Maycock, T. Waterfield, O. Yelekçi, R. Yu, and B. Zhou (eds.)]. Cambridge University Press, Cambridge, United Kingdom and New York, NY, USA, pp. 287–422, doi:10.1017/9781009157896.004

Szopa, S., V. Naik, B. Adhikary, P. Artaxo, T. Berntsen, W.D. Collins, S. Fuzzi, L. Gallardo, A. Kiendler-Scharr, Z. Klimont, H. Liao, N. Unger, and P. Zanis, 2021: Short-Lived Climate Forcers. In Climate Change 2021: The Physical Science Basis. Contribution of Working Group I to the Sixth Assessment Report of the Intergovernmental Panel on Climate Change [Masson-Delmotte, V., P. Zhai, A. Pirani, S.L. Connors, C. Péan, S. Berger, N. Caud, Y. Chen, L. Goldfarb, M.I. Gomis, M. Huang, K. Leitzell, E. Lonnoy, J.B.R. Matthews, T.K. Maycock, T. Waterfield, O. Yelekçi, R. Yu, and B. Zhou (eds.)]. Cambridge University Press, Cambridge, United Kingdom and New York, NY, USA, pp. 817–922, doi:10.1017/9781009157896.008

Wang, H., Lu, X., Jacob, D. J., Cooper, O. R., Chang, K.-L., Li, K., Gao, M., Liu, Y., Sheng, B., Wu, K., Wu, T., Zhang, J., Sauvage, B., Nédélec, P., Blot, R., and Fan, S. (2022), Global tropospheric ozone trends, attributions, and radiative impacts in 1995–2017: an integrated analysis using aircraft (IAGOS) observations, ozonesonde, and multi-decadal chemical model simulations, Atmos. Chem. Phys., 22, 13753–13782, https://doi.org/10.5194/acp-22-13753-2022

Wasserstein, R. L., Schirm, A. L., and Lazar, N. A.: Moving to a world beyond p < 0:05, Am. Stat., 73, 1–29, https://doi.org/10.1080/00031305.2019.1583913, 2019.

---

## Author Response (AR1)

**Author Responses to Reviewer Comments**

We thank the reviewers for their useful and constructive comments/feedback. We also thank Owen Cooper for his useful comments on our manuscript in relation to the TOAR-II special edition. We have reproduced their comments below in black text, followed by our responses in red text. Please note, we have number listed Reviewer #2's and Owen Cooper's comments for clarification when addressing comments relevant to both reviewers. Any additions to the manuscript are in blue text and our reference to line numbers is based on the originally submitted manuscript.

**Reviewer 1's Comments:**

This manuscript provides a global assessment of the lower-tropospheric ozone (LTCO3) retrievals from four satellite instruments, mainly focusing on vertical sensitivities and long-term stabilities. It then goes on to create a harmonized long-term (1996-2017) time series of LTCO3, from which inter-annual variations in global LTCO3 are evaluated. As the first effort to investigate this issue, this paper is well-designed and does not exhibit obvious flaws. The topic is significant considering the stronger relevance of LTCO3 with anthropogenic activities and ozone radiative forcing than tropospheric ozone. Certain details are lack in the present form of this manuscript, which I encourage the authors to elaborate more during the revision. I support the publication of this paper on ACP if the following comments can be addressed.

1) One paragraph of discussing why LTCO3 warrants investigation (relative to tropospheric ozone) should be added to the Introduction Section. For example, LTCO3 (if detectable) can be more relevant with surface ozone pollution and anthropogenic ozone radiative forcing.

We have added the following paragraph after Line 71:

"While many studies have previously focussed on $TCO_3$ (e.g. Gaudel et al. (2018); Ziemke et al. (2019)), several nadir-viewing ultraviolet-visible (UV-Vis) sounders can retrieve $TO_3$ between the surface to 450 hPa (i.e. lower tropospheric column $O_3$, $LTCO_3$). The retrieval scheme from the Rutherford Appleton Laboratory (RAL) Space exploits information from the $O_3$ Huggins bands (325-335 nm), as well as the Hartley band (270-307nm), to retrieve high quality $LTCO_3$ and was selected for the ESA-CCI and EU Copernicus Climate Change Service. As a result, the RAL Space $LTCO_3$ products (and equivalent from other providers) are valuable resources to investigate global and regional $O_3$-related air quality (e.g. Richards et al., 2013; Pope et al., 2018; Russo et al., 2023).".

We have added the following reference for Russo et al., (2023):

Russo, M.R., Kerridge, B.J., Abraham, N.L., et al. 2003. Seasonal, interannual and decadal variability of tropospheric ozone in the North Atlantic: comparison of UM-UKCA and remote sensing observations for 2005-2018. *Atmospheric Chemistry and Physics*, **23** (11), 6169-6196, doi: 10.5194/acp-23-6169-2023.

The paragraph starting on Line 72 has been modified slightly to account for some acronyms being defined in the new paragraph above:

"In this study, we explore the spatiotemporal variability of $LTCO_3$ from several UV-Vis sounders produced by RAL Space.".

2) The satellite sensitivity to LTCO3 (DOFs) varies spatially, especially following the distribution of LTCO3 abundances. So hemispheric averaged DOFs in Table 2 cannot provide a comprehensive enough insight of this sensitivity. I suggest to additionally include a map or histogram of DOFs (associated with Figure 3) of each instrument. Another issue to further investigate is: should we also consider DOFs at each location when evaluating the significance of derived trends in LTCO3? Are DOFs too small in some "green polygons" in Figures 6 and 7?

We point the reviewer to our response to the comments from Reviewer #2 comment #2.

3) Figure 2: in many cases, LTCO3 from Ozone Sondes and a priori are close to each other, while applying the AKs (satellite and sonde-AK) modifies them into different (usually opposite) directions (e.g., in the OMI global case). Should provide an explanation on this issue.

The prior profile is based on a climatology (McPeters et al., 2007) which is itself derived from a multi-year ozonesonde ensemble. The ozonesonde and a priori are therefore expected to be close. The sign of the change from sonde to sonde-adjusted for averaging kernel and prior depends on both aspects. The adjusted sonde will move further away from the prior than the unadjusted sonde if the influence of higher layers on the lower troposphere layer retrieval is large enough and pulls in that direction. It should also be noted that while intuitively vertical smearing from the stratosphere would be expected to increase the tropospheric layer retrieval (so AK adjustment would decrease the sonde value), in the case of OMI there is a negative excursion in the AK in the lowest stratosphere, so the opposite is true.

To clarify this in the manuscript we have added on Page 5 Line 173:

"This increase in LTCO$_3$ when the OMI AKs are applied to the ozonesondes contrasts with the other satellite instruments. While the vertical smearing from the stratosphere would intuitively be expected to increase the tropospheric layer retrieval, and thus the AK adjustment to decreases the ozonesonde value, in the case of OMI there is a negative excursion in the AKs into the lowermost stratosphere (see **Figure 1**), so the opposite occurs.".

The McPeters et al., (2007) reference is below.

McPeters, R.D., Labow, G.J., and Logan, J.A. 2007. Ozone climatological profiles for satellite retrieval algorithms. Journal of Geophysical Research, 112 (D05308), https://doi.org/10.1029/2005JD006823.

4) Figure 3: The increasing trends in LTCO3 (as discussed later in the paper) might be responsible for some differences between instruments averaged for varying periods. This fact and the expected impacts on the inter-comparison here should be discussed.

This is a good point. We have added the following sentence on Page 5 Line 195:

"These differences in the GOME-1 and GOME-2/OMI LTCO$_3$ seasonal averages are likely to be at least partly due to underlying LTCO$_3$ tendencies between the respective instrument time periods. This is investigated further in Section 3.4.".

5) Line 240-243: More details should be provided to justify the choice to only drop 2003 in the merging for GOME-1, since 2002 seems also to potentially introduce artificial trends.

The reviewer is correct that, in Figure 4 in the tropics, the year 2002 does have a low bias like 2003. Having said that, the SH biases are small and the NH biases are lower in 2003 (~-5.0 DU) than 2002 (~-3.0 DU). Therefore, 2002, on the whole, does not seem as affected as does 2003. Also, both GOME-1 time-series in Figure 5 show similar values for 2000 to 2002, so 2002 does not appear to be substantially lower than neighbouring years. Finally, as SCIAMACHY does not have a full year of data for 2002, there is no full data set to gap-fill 2002. As a result, with the data sets available, we are confident in the harmonised data set we have created. To make it clearer why we have included 2002 in our harmonised time-series, we have added on Line 263:

"While GOME-1 in the tropics (**Figure 4**) for 2002 has a negative bias (~5.0 DU) in comparison to the ozonesondes, the biases for the other latitude bands are less distinct. The regional average $LTCO_3$ values for 2002 in **Figure 5** are also comparable to neighbouring years (e.g. 2000 and 2001). SCIAMACHY also does not have a full year of data for 2002, so we have included the GOME-1 2002 data in our analysis.".

6) Figure 1 is unnecessary (at least in the main text) considering more clear and relevant information is already available in Table 2.

We agree with Reviewer #1 that we do not need Figure 1 and Table 2. However, we are more inclined to remove Table 2 and leave Figure 1 as it provides useful information on the structure of the AKs for the different instruments. However, we have updated Figure 1 in line with Reviewer #2's comment #2.

**Reviewer 2's Comments:**

In this work, Pope et al. provide an "Investigation of spatial and temporal variability in lower tropospheric ozone from RAL Space UV-Vis satellite products" that is suitable for publication in the TOAR-II special issue. Given their extensive expertise with the data, however, the authors sometimes seem to be too brief/quick in passing their message and conclusions to a possibly less-informed audience. Therefore, the following clarifications/improvements are suggested:

1. Lines 15, 67, 135, 137, 178-180: In contrast with common practice based on the work of Rodgers (2000), the terms (vertical) sensitivity and degrees of freedom are both used without distinction. Whereas the former rather refers to the integrated weighting function (averaging kernel row sum) as a measure of the fraction of the info that comes from the retrieval, the latter refers to the kernels' diagonal elements only as an indication of the number of independent layers that is retrieved. Ideally, the authors would address both, though clearly distinguished, in their discussion. Other retrieval performance measures like the effective vertical resolution of the LTCO3 layer and the retrieval weight offset would be most helpful in the discussion, but might not be mandatory.

On Page 2 Lines 67-69 we have already defined the link between the vertical sensitivity and the AK:

"The vertical sensitivity can be referred to as the "averaging kernel" (AK), which provides the relationship between perturbations at different levels in the retrieved and true profiles."

For the definition of the DOFS on Page 4 Line 135 we have added the sentence:

"Here, the DOFS represents the number of independent pieces of information on the vertical profile in the retrieval (i.e. the sum of the AK diagonal).".

In terms of other performance metrics, as suggested by Reviewer #2 comment #2, we have added seasonal maps (i.e. new Figure 3) of the DOFS for the LTCO$_3$.

2. Relatedly, for Figure 1: Is this for all retrieved satellite pixels within 90° latitude bands? For clarity, possibly provide the effective latitude range (and number of averaged observations) for each month given the lack of observations during polar night? Additionally, it would be helpful to specify whether the lowest red curve indeed corresponds to the LTCO3 retrieval, and to what extent this kernel agrees with or differs from ideal AK behavior (for results discussion). On the other hand, for a full and proper interpretation of the results in Figs. 6 and 7, it would be appropriate to have a view on the DFS (or integrated sensitivity) on the same spatial resolution (latitude-longitude boxes) as Figs. 6 and 7), or at least the spatiotemporal resolution of Fig. 4. The shifts in spatiotemporal resolutions between all consecutive figures (except for 6 and 7) seems to hamper a fully correlative interpretation.

Reviewer #1 made a similar point in their comment #6 stating that Figure 1 was unnecessary given the information in Table 2. However, we prefer to keep Figure 1 as it provides useful information on the structure of the AKs and there are subtle differences between hemispheres, months and instruments. However, in line with Reviewer's #2 comment, we have restricted the AK plot to 60°S&N to avoid the polar night (i.e. similar latitude range to Figures 6 and 7). We have also added N= in each panel to state the number of retrievals used in the average AK plot.

We have also taken this a step further by adding a new Figure 2 showing the spatial distribution of the seasonal DOFS. We have also regridded the horizontal resolution of Figure 3 to that of Figure 6 and 7 for consistency, as suggested by Reviewer #2. The new Figure 2 and Figure 1 for the DOFS are presented below with updated/new manuscript text. We have also deleted the sentence "*The OMI, GOME-1, GOME-2 and SCIAMACHY level 2 data were aggregated on a 1.0°×1.0° spatial grid using the gridding approach of Pope et al., (2018).*" from Page 3 Lines 98-99.

[Figure]

*Figure 1*: *Average averaging kernels (AKs) for the instruments listed in **Table 1** for the northern and southern hemispheres (60°S-60°N) in January and July of 2008 (1998 for GOME-1). The average degrees of freedom of signal (DOFS) is shown as is DOFS LTCO₃ which represents the DOFS in the lower tropospheric column ozone (LTCO₃). N represents the number of retrievals in each average AK average.*

[Figure]

*Figure 2: Seasonal distributions of LTCO$_3$ degrees of freedom of signal (DOFS) in DJF and JJA for GOME-1, GOME-2, OMI and SCIAMACHY averaged over the full record for each instrument.*

On Page 4 Line 144 we have added discussion on the new Figure 2:

"While **Figure 1** provides spatial average information on LTCO$_3$ DOFS, **Figure 2** shows spatial maps for December-January-February (DJF) and June-July-August (JJA) over the respective instrument records. The largest LTCO$_3$ DOFs occur over the ocean ranging between approximately 0.4 and 0.6 for GOME-1,

GOME-2 and SCIAMACHY, while OMI has larger ocean values between 0.7 and 0.8. Over land, the $LTCO_3$ DOFS tend to be lower and between 0.3 and 0.5 for GOME-1, GOME-2 and SCIAMACHY. Again, OMI has large values on land of between 0.4 and 0.7. Depending on the hemispheric season, the summer-time (JJA in NH and DJF in SH) $LTCO_3$ DOFS are larger for each instrument. Overall, OMI (GOME-2) retrievals contain the largest (lowest) information on $LTCO_3$.".

We have also added on Page 7 Line 280:

"As shown in **Figure 2**, there is sufficient information (e.g. $LTCO_3$ DOFS mostly > 0.5) in the tropics and mid-latitudes for the instruments used to form the merged $LTCO_3$ data. This provides confidence in this merged $LTCO_3$ record for long-term temporal analysis.".

For the new Figure 1, the text on Page 4 Lines 132-143 has been updated to reflect some changes to the figure:

"**Figure 1** represents average AKs for all the instruments listed in **Table 1** for 2008 (1998 for GOME-1) in the northern (NH) and southern (SH) hemispheres between the equator and 60°S & N. Of the four RAL Space products, OMI $O_3$ profiles appear to contain the most information with degrees of freedom of signal (DOFS) of 5.0 or above for the full atmosphere. SCIAMACHY has the lowest sensitivity with average DOFS ranging between 4.12 and 4.64. The DOFS tends to be larger in NH for all the products, though there is no clear pattern in the seasonality (i.e. January vs. July). In terms of $LTCO_3$, OMI again has greater sensitivity than the others with average hemispheric and seasonal DOFS ranging between 0.63 and 0.68. For GOME-1 (GOME-2), the $LTCO_3$ DOFS range between 0.37 and 0.50 (0.39 and 0.46). SCIAMACHY $LTCO_3$ DOFS range between 0.44 and 0.52. Therefore, while SCIAMACHY has the lowest overall sensitivity to full atmospheric ozone, it has reasonably good information in the $LTCO_3$, as do the other instruments. These results are robust given the large number of retrievals (N) that have been used to derive the average AKs (i.e. N > 65,000 in all cases).".

Also, on Page 5 Lines 186-187, we have replaced "*Here, we compare the long-term seasonal (December-January-February, DJF, and June-July-August, JJA) spatial distributions of RAL Space $LTCO_3$ products (Figure 3).*" with "Here, we compare the long-term seasonal (DJF and JJA) spatial distributions of RAL Space $LTCO_3$ products (**Figure 3**).".

The text relating to Figure 2 in Section 3.1 has now been updated to discuss Figure 3 with the addition of the new figure. On Page 4 Line 144-145, the text "*This is investigated further by co-locating the products with the merged ozonesonde data set, over their respective mission periods, globally and in the NH and SH (Figure 2).*" has been updated to "The impact of the satellite vertical sensitivity is further investigated by co-locating the products with the merged ozonesonde data set, over their respective mission periods (globally and in the NH and SH) and the AKs applied to assess the impact on the ozonesondes (**Figure 3**).".

3.  Lines 53-56 and 62-64: The inconsistencies between the instruments as described in Gaudel et al. (2018) also originate from different prior information choices, and different top-level definitions for vertical integration and ozone burden calculations.

To address this comment, we have added the following sentence on Line 64: "Another factor introducing inconsistencies is the assumed tropopause height for the different products. Some products used the World Meteorological Organisation (WMO) definition of "the first occurrence of the 2 K/km lapse-rate" while some others integrated the 0-6 km and 6-12 km sub-columns to derive the tropospheric column. The use of different a priori products within the retrieval scheme will have also provided inconsistencies.".

4. Lines 72-73: "In this study, we explore the spatiotemporal variability of lower tropospheric column ozone (LTCO3, surface to 450 hPa)…" Please explain the choice of this fixed pressure level, in terms of the fixed (?) retrieval grid and vertical sensitivity (stratospheric info permeating the tropospheric retrieval output, also see first comment).

This pressure range is based on the legacy of the RAL retrieval scheme which has been used in previous studies (e.g. Munro et al., 1998 – doi:10.1038/32392; Miles et al., 2015 - doi:10.5194/amt-8-385-2015). As demonstrated in this and earlier work (e.g. Russo et al (2023) - 10.5194/acp-23-6169-2023 and Miles et al., (2015)), this retrieval setup derives robust estimates of lower tropospheric ozone between the surface and 450 hPa. As for the "stratospheric info permeating the tropospheric retrieval output" comment, we have addressed this in response to Reviewer #2's comment #1.

5. Section 2.1 and Table 1: What is now in a footnote sounds too important not to be included in the main text. The main differences between the product versions are expected to be explained, possibly summarized as different parameter settings in Table 1 (instead of the product link for example, which is the same for all and hence superfluous as a separate column). The conclusions on the LTCO3 differences between the products are expected to relate to this then.

The main difference between the fv300 and fv214 retrieval versions is that GOME-1, GOME-2 and SCIAMACHY are across scanning instruments while OMI has a 2-D array detector. Therefore, there are subtle differences in the retrieval scheme versions to account for this. We have since removed the footnote and added on Page 3 Line 95:

"The differences between the retrieval versions (i.e. fv214 and fv300) in **Table 1** are primarily linked to the instrument types where GOME-1, GOME-2 and SCIAMACHY are across-track scanning instruments while OMI uses a 2-D array detector.".

6. Lines 117-118: Are all co-locations considered in the analysis, or only the closest in space or time? Also possibly provide more information or reference(s) on "representation errors". Something like "spatiotemporal sampling difference errors" may sound clearer.

It is only the satellite retrieval which is co-located most closely in time and space with the ozonesonde profile. To make this clearer, we have reworded Lines 117-118:

"In this study, for comparisons between ozonesonde profiles and satellite retrievals, each ozonesonde profile was spatiotemporally co-located within 500 km and 6 hours to allow for robust comparisons and reduce representation errors." to

"In this study, for comparisons between ozonesonde profiles and satellite retrievals, each ozonesonde profile was spatiotemporally co-located to the closest satellite retrieval. Here, all the retrievals within 6

hours of the ozonesonde launch were subsampled and then the closest retrieval in space (i.e. within 500 km) was taken for the final co-located one. Therefore, there was one satellite retrieval for every ozonesonde profile to help reduce the spatiotemporal sampling difference errors.".

7. Line 128: "sonde sub-column profile (DU) on the satellite pressure grid" The sonde data are typically not provided as sub-columns, and certainly not on the satellite pressure grid. Please provide a reference and/or some clarification for the conversion of the initial sonde data to this derived product.

To clarify this we have added on Page 4 Line 129:

"Here, the ozonesonde profile, on its original pressure grid (typically in units of ppbv or mPa) are converted into ozone sub-columns between each pair of levels. These sub-columns are then aggregated up to the larger sub-columns (e.g. the $LTCO_3$ range is between the surface and 450 hPa) on the coarser satellite pressure grid.".

8. Section 3.2: Do these results agree with the ozonesonde comparisons then? Please briefly discuss.

Yes, there is agreement in the seasonality between the satellite and ozonesondes $LTCO_3$ values. Using the ozonesonde data between 1996 and 2017 we derived hemispheric DJF and JJA median (25$^{th}$ and 75$^{th}$) $LTCO_3$ values. For the northern hemisphere in DJF and JJA, the $LTCO_3$ values were 18.0 (15.7, 20.0) DU and 20.8 (16.7, 24.6) DU, respectively. For the southern hemisphere in DJF and JJA, the $LTCO_3$ values were 10.8 (8.2, 14.8) DU and 14.4 (12.1, 16.3), respectively. We have added the following paragraph on Page 5 Line 205:

"The satellite $LTCO_3$ seasonality is consistent with that of the ozonesondes. Here, the median (25$^{th}$ percentile, 75$^{th}$ percentile) ozonesonde $LTCO_3$ values for the NH in DJF, NH in JJA, SH in DJF and SH in JJA are 18.0 (15.7, 20.0) DU, 20.8 (16.7, 24.6) DU, 10.8 (8.2, 14.8) DU and 14.4 (12.1, 16.3) DU, respectively. Therefore, the NH $LTCO_3$ values are larger than those in the SH and the JJA $LTCO_3$ values are larger than the DJF equivalent. All of which are consistent with the four instrument $LTCO_3$ seasonal distributions.".

We have also exploited the ozonesonde data to look at the 1996-2000 average vs. 2013-2017 average differences in the old Figure 7. Here, we calculated the differences at ozonesonde sites which had annual average data for at least 75% of the 22-years between 1996 and 2017. The new/updated Figure 7 is below. We have added the following text on Page 8 Line 308:

"The ozonesondes are consistent with satellite 1996-2000 and 2013-2017 average $LTCO_3$ differences. In the tropics, the majority of ozonesonde sites show increases between these two periods ranging between 0.5 and 5.0 DU. Over Europe (i.e. northern mid-latitudes), the ozonesonde $LTCO_3$ differences range between -0.5 and 0.5 DU suggesting limited $LTCO_3$ change over time.".

[Figure]

**Figure 7**: LTCO3 (DU) merged data set from GOME-1 (1996-2002), SCIAMACHY (2003-2004) and OMI (2005-2017) where the difference between the 2013-2017 average and 1996-2000 average is shown. Green polygon-outlined regions show substantial differences (95% confidence level and where the absolute difference > 1.0 DU) using the Wilcoxon Rank Test. Grey pixels are where the South Atlantic Anomaly influence on retrieved $LTCO_3$ has been masked out. Circles show differences in ozonesonde $LTCO_3$ (DU) over the same time periods as the merged satellite record.

9. Lines 265-266: "Based on the difference between 2002 and 2005, a global scaling is applied in 2003 and 2004 for the SCIAMACHY spatial fields." Does this mean that the scaling changes linearly in time, with yearly resolution?

Yes, the reviewer is correct. Unfortunately, the GOME-1 data for 2003 and 2004 is not suitable for use in the harmonised time series (i.e. Lines 238 – 243). Therefore, for these two years, we adopt the SCIAMACHY spatial pattern but use the global $LTCO_3$ scaling linearly interpolated in time per year to scale the SCIAMACHY data in line with that of GOME-1 2002 and OMI 2005. To make this clearer, we have updated the following text on Lines 265-266 from:

"Based on the difference between 2002 and 2005, a global scaling is applied in 2003 and 2004 for the SCIAMACHY spatial fields." with "Based on the difference between 2002 and 2005, an annual linear global scaling is applied in 2003 and 2004 for the SCIAMACHY spatial fields.".

10. Line 296: "Overall, these anomalies suggest there has been limited change in LTCO3, between 1996 and 2000" This cannot be seen from Fig. 6. Rather immediately refer to Fig. 5, as is done on the next line. Together with that, it would seem appropriate to also discuss the decrease in LTCO3 towards the end of the time series, from about 2013-2014 onwards.

This was a mistake and "between 1996 and 2000" should have been "between 1996 and 2017". This has now been updated.

11. Lines 320-323: The study of Wespes et al. (2018) comes with some caveats that should be mentioned for proper interpretation. More importantly, this does not straightforwardly allow concluding that all "studies using IR products tend to show significant negative trends globally, while studies using UV-Vis products show significant increasing trends in the tropics/sub-tropics." See also comment on lines 53-56 and 62-64, which could then be specified with "other improvements" on line 339.

To account for this, we have updated the text on Page 8 Lines 319-323:

"However, the study by Wespes et al., (2018) indicates that TCO3 has been significantly decreasing between 2008 and 2017 at -0.5 to -0.1 DU/yr from IASI (i.e. an IR sounder). Therefore, studies using IR products tend to show significant negative trends globally, while studies using UV-Vis products show significant increasing trends in the tropics/sub-tropics." to

"However, the study by Wespes et al., (2018) from IASI (an IR sounder) indicated that $TCO_3$ decreased between 2008 and 2017 by -0.5 to -0.1 DU/yr. Gaudel et al., (2018) reported similar $TCO_3$ tendencies using two IASI products (IASI-FORLI and IASI-SOFRID). However, Boynard et al., (2018) and Wespes et al., (2018) report a step-change in 2010 in the IASI-FORLI ozone data which could influence observed long-term trends. Therefore, studies using IR products available to TOAR-I and Wespes (2018) are no longer considered reliable.".

Technical corrections:

Lines 29-30: Providing a maximum does not say anything about the retrievals in general. This statement is too optimistic.

We point the reviewer to our response to Reviewer #2's comment #2.

Lines 40-41: Add whether this conclusion agrees with or contrasts common literature?

This is slightly more problematic to address given that we are the first study to look at $LTCO_3$ specifically and its temporal evolution using a harmonised product. In Section 5: Discussion and Conclusions we discuss the trends in tropical $TCO_3$ and state the similarity in our $LTCO_3$ and suggest they are likely linked (i.e. trends in $LTCO_3$ are substantially contributing to the $TCO_3$ trends). Therefore, on Lines 40-41, we have updated "Therefore, we conclude that there has been a substantial increase in tropical/sub-tropical LTCO3 during the satellite-era." with "Therefore, we conclude that there has been a substantial increase in tropical/sub-tropical $LTCO_3$ during the satellite-era, which is consistent with tropospheric column ozone ($TCO_3$) records from overlapping time-periods (e.g. 2005-2016).".

Line 57: "one" to "on one"

This has been updated.

Line 60: remove "that there is"

This has been removed.

Line 70: "so might the processes controlling variability in retrieved TO3" is not very clear. Please extend/rephrase.

We have rephrased "As the instruments' vertical sensitivities differ so might the processes controlling variability in retrieved $TO_3$ and so trends may also differ between products." with "As the instruments' vertical sensitivities differ, they are likely to be influenced differently by processes controlling $TO_3$ temporal variability in different layers of the troposphere (e.g. lower troposphere influenced more by precursor emissions vs. the upper troposphere subject more to influence from stratospheric-tropospheric exchange). Therefore, the differing vertical sensitivities, and thus the $TO_3$ they are retrieving, could be driving the inconsistencies in reported $TCO_3$ trends between products.".

Line 78: rephrase "presents are results"

We have reworded "In our manuscript, section 2 discusses the satellite/ozonesonde datasets, section 3 presents are results and our conclusions/discussion are summarised in section 4." with "In our manuscript, Section 2 discusses the satellite/ozonesonde datasets used, Section 3 presents are results, while Section 4 summarises our conclusions and discussion points.".

Line 92: remove "ozone"

This has been removed.

Line 112: The IASI-FORLI products are not in Table 1.

We have updated "products of the RAL and IASI-FORLI product listed in **Table 1**" with "products from RAL listed in **Table 1**".

Line 176: rephrase "is remains"

Have deleted the "is".

Line 207: rather "corrected for" than "tolerated" ?

We prefer our terminology in this instance.

Figure 4: Mention latitude ranges for different drift results in caption.

On 492 we have added "The latitude bands are 90-30°S, 30°S-30°N and 30-90°N.".

Figure 5: Is there a reason for not mentioning SCIAMACHY in the caption for the 2003-2005 period?

On Line 510 we have added "The merged record also includes globally scaled $LTCO_3$ data from SCIAMACHY for 2003 and 2004.".

Line 281: "similar"

This has been corrected.

Line 287: "Pirovano et"

This has been corrected.

Line 289: rephrase "There are a…"

One Line 289 this has been reworded to "There are small clusters of substantial anomalies…."

**Owen Cooper's Comments:**

1. Discussion of trends: The expression "statistically significant" is used throughout the submitted manuscript, however this expression is now recognized as being problematic and it should be abandoned and replaced by the more useful method of reporting all trends (with uncertainty, e.g. 95% confidence intervals) and all p-values, followed by a discussion of the trends and the author's opinion regarding their confidence in the trend values. This advice comes from a highly influential paper by Wasserstein et al. (2019), published in the journal, The American Statistician, that has already been cited over 1400 times (according to Web of Science). This advice was adopted by the first phase of TOAR (Tarasick et al., 2019) and will also be used by TOAR-II. Some other recent papers on ozone trends that have taken this advice are: Chang et al., 2020; Cooper et al., 2020; Gaudel et al., 2020; Chang et al., 2022; Wang et al., 2022. Because these papers report all trend values, uncertainties, and all p-values, and also discuss the trend results, there is no confusion regarding the findings, and one does not even notice that the term "statistically significant" is not used at all. Table 3 of the TOAR-II statistical guidelines provides calibrated language for describing trends and uncertainty, similar to the approach of IPCC.

In line with the collective comments from Owen Cooper, we have added a new Table (see Owen Cooper's comment #2) to show the linear LTCO$_3$ trends in 15° latitude bands between 60°S-60°N with the corresponding p-values and 95% confidence intervals. However, our view continues to be that the 5-year anomaly maps provide an important visual aspect to the key result from this paper. It provides a clear view of the spatial changes in LTCO$_3$ over the merged instrument record. The polygon-outline regions also provide useful information on where there are "substantial differences" in the climatological median value and 5-year median value (i.e. where the p-value is <0.05). Therefore, we retain the 5-year map analysis but in line with Owen Cooper's suggestions, we have reworded "statistically significant" replacing it with language such as "substantial differences".

2. Below is a figure from Chapter 2 of IPCC AR6 WG-I (Gulev et al., 2021) summarizing observed global ozone trends. TOAR-II will produce a similar figure from all recent ozone trend studies published in the TOAR-II Community Special Issue (as well as from studies not in the special issue). Trends from all new satellite studies can be added to the right-hand panel, but the trends must be reported in units of ppbv/decade, with p-values and 95% confidence intervals. Your study currently reports ozone changes in Dobson units from one 5-year period to the next. Could you report these values as trends in units of ppbv/decade? You can choose whichever latitude bands you like when reporting the zonal trends, but useful intervals would be 10 or 15 degrees.

As mentioned in our response to Owen Cooper's comment #1, we have added a new Table and sub-section in Section 3 to discuss zonal trends in $LTCO_3$. Here, for the latitudinal trends, we have the values as DU/decade and ppbv/year with p-values and 95% confidence ranges.

| Latitude Band | LTCO$_3$ Trend (DU/decade) (95% Confidence Interval) | LTCO$_3$ Trend (ppbv/decade) (95% Confidence Interval) | p-values |
|---|---|---|---|
| 60°S ≤ Latitude < 45°S | 4.49 (2.51, 6.48) | 10.37 (5.79, 14.95) | 0.00 |
| 45°S ≤ Latitude < 30°S | 1.85 (0.11, 3.59) | 4.27 (0.26, 8.28) | 0.03 |
| 30°S ≤ Latitude < 15°S | 0.94 (-1.05, 2.93) | 2.17 (-2.42, 6.76) | 0.35 |
| 15°S ≤ Latitude < 0° | 2.89 (1.27, 4.52) | 6.68 (2.94, 10.43) | 0.00 |
| 0° ≤ Latitude < 15°N | 3.93 (3.13, 4.72) | 9.06 (7.23, 10.89) | 0.00 |
| 15°N ≤ Latitude < 30°N | 4.12 (3.25, 4.97) | 9.50 (7.51, 11.48) | 0.00 |
| 30°N ≤ Latitude < 45°N | 1.33 (-0.34, 3.01) | 3.08 (-0.78, 6.95) | 0.11 |
| 45°N ≤ Latitude < 60°N | 0.49 (-1.14, 2.13) | 1.14 (-2.64, 4.91) | 0.55 |

**Table 2**: LTCO$_3$ trends (DU/decade and ppbv/decade) for latitude bands (15° bins) between 60°S and 60°N. The 95% confidence intervals of the trends are shown in brackets. The trend p-values are also shown.

We have added the following sub-section (note Figure 7 has been updated Figure 8 with the new DOF figure):

**3.6. Long-term LTCO$_3$ Trends**

"In line with TOAR-II, we have added additional metrics on the temporal change in LTCO$_3$ over the merged instrument record. Here, we have calculated the linear trends in LTCO$_3$ in 15° latitude bins between 60°S-60°N along with the 95% confident range and associated p-values (see **Table 2**). In the tropical latitudes (15°S-30°N), all the linear trends show substantial increasing trends (2.89-4.12 DU/decade) between 1996 and 2017; all with p-values tending to 0.0. This is consistent with the LTCO$_3$ positive differences (3.0-5.0 DU) between the 1996-2000 and 2013-2017 averages (**Figure 8**). In the northern mid-latitudes (30-60°N), there are smaller positive trends (1.33 and 0.49 DU/decade) but the 95% confidence values intersect with 0.0 and have larger p-values. Again, this is consistent with the near-zero differences between the 1996-2000 and 2013-2017 averages (**Figure 8**). In the southern mid-latitudes (30-60°S), the trends are substantially positive (1.85 and 4.49 DU/decade) with near-zero p-values. Again, this is consistent with the substantial differences (2.0-4.0 DU) between the 1996-2000 and 2013-2017 averages. The 15-30°S trend is small at 0.94 DU/decade with a moderate p-value of 0.35, indicating this not to be significant.".

3. This paper reports satellite ozone retrievals for the lower troposphere, which is defined as the surface to 450 hPa. Typically, when one thinks of the lower troposphere, the layer from the surface to about 700 or 600 hPa comes to mind, for example the IASI-GOME2 product focuses

on the lower troposphere and spans the layer from the surface to 3 km. The layer from the surface to 450 hPa also includes the region of the atmosphere that is typically thought of as the mid-troposphere (600-400 hPa). In cold months at high latitudes the tropopause is often as low as 300 hPa, in which case the surface-450 hPa layer spans most of the troposphere. To avoid confusion with other products/definitions, and to highlight that the new RAL product also spans the mid-troposphere, can the product be referred to as lower-mid tropospheric? Line 321 The paper sates that IR satellite instruments tend to report negative ozone trends globally, but I don't think that such a generalization can be made.

Overall, we are inclined to retain the definition of $LTCO_3$ for our analysis. Figure 3 from the originally submitted manuscript clearly shows spatial signals of ozone originating from the surface/boundary layer. The AKs in Figure 1 from the originally submitted manuscript also show peak sensitivity in lower troposphere (i.e. below 700 hPa). Therefore, the retrieved ozone signal in this sub-column emanates from the lower troposphere and thus we retain the current definition.

In response to the comment "The paper states that IR satellite instruments tend to report negative ozone trends globally, but I don't think that such a generalization can be made.", we addressed this in response to Reviewer #2's comment #11.

4.  Gaudel et al (2018) compared two IASI ozone products and the trends don't always agree. As shown in Figure 23, In the northern tropics IASI-FORLI shows a weak positive trend (IASI-SOFRID is negative), while at northern mid-latitudes IASI-SOFRID shows no trend but IASI-FORLI is strongly negative. After the publication of Gaudel at al. (2018), Boynard et al. (2018) reported a negative drift in the IASI-A instrument. If this drift is taken into account in updated IASI trends then the negative trends may not be so strong.

We addressed this in response to Reviewer #2's comment #11.

5.  Ziemke et al. (2019) found a positive drift in the OMI/MLS product and added a correction to their final product to account for the drift. The submitted paper reports drift values for all products but only found the GOME-II drift to be of concern and rejected that particular dataset. The drift for OMI was reported as 0.22 DU/yr in the tropics, but deemed insignificant. But still, after 15 years the drift would add up to 3.3 DU, which is more than 10-20% of the tropical tropospheric ozone column. Was a correction applied to account for drift?

Analysis from the RAL Space team of OMI data produced using their retrieval scheme over an extended period indicates that bias in input UV reflectance data occurs mainly in years more recent than those used in our analysis (i.e. after 2017). Therefore, we are confident in the approach we have used to avoid possible effects of OMI instrument degradation. The data filtering stated in Section 2.1 aims to remove retrieved $LTCO_3$ values that are not suitable for use in scientific analysis (i.e. not of sufficient quality). Pixels/rows substantially influenced by the so-called OMI row anomaly drop out as the record progresses and thus the across-track sampling drops. This can be seen from **Figure R1,** which shows the number of $LTCO_3$ retrievals for each OMI row (note there are 60 OMI rows but to improve the signal-to-noise-ratio in the RAL OMI fv214 product, pairs of rows were combined when retrieving $LTCO_3$) per year in several latitude bands. Here, the number of retrievals per year is >500,000 at the start of the timeseries (e.g. round 2005-2009) and then decreases to approximately >250,000 retrievals per year at the end (e.g. 2015-2017) for some latitude bands/pixels. White regions show where there is no LTCO$_3$ data for that OMI row for that year e.g. all data for row 29 drops off after 2009. However, there is still a large proportion of data per year per latitude band (i.e. several million) meaning that there is sufficient data for our study.

[Figure]

**Figure R1**: Frequency distribution of OMI fv214 LTCO$_3$ retrievals in each OMI row-time bin for the latitude bands: 30-60°S, 0-30°S, 0-30°N and 30-60°N.

Secondly, the satellite-ozonesonde bias trend in the tropics (0.22 DU/yr – **Figure 4**) had a p-value of 0.0103. Here, the biases are typically -5.0 to -3.0 DU in 2005-2007, tend to near zero in 2010-2012 and decrease again to -2.0 to -1.0 DU in 2015-2017. Therefore, there is a non-linear pattern in the bias and yes, the linear trend is positive, but robustness of the value is limited by the bias variability in the OMI record. Therefore, if the OMI row anomaly, as expected, would yield a deterioration in the OMI LTCO$_3$ with time, there would be a linear bias propagating in time, which is not observed. Secondly, when the merged data set is derived, the GOME-1 bias-corrected and OMI timeseries between 2005 and 2010 are averaged together as shown by the black line in Figure 5. Therefore, this is likely to outweigh an underlying trend in the OMI data due to the row anomaly. It is also worth pointing out that the GOME-1 biases with the ozonesondes (i.e. 1996-2000) in the tropics are near-zero while the OMI biases between 2013 and 2017 are slightly negative. Therefore, based on the satellite-sonde biases between the start and end of the record, the merged LTCO$_3$ should have a negative bias trend which is likely slightly buffering the reported positive differences in Figure 7 in the tropics.

As a result of the number of retrievals still used in the OMI record through time and the caveats with the OMI-ozonesonde tropical bias trend of -0.22 DU/year, we are confident we have accounted for, or mitigated against, any influence the OMI row anomaly has on the temporal changes reported in this study. However, to make this clear to the reader we have added on Page 3 Line 99:

"These filters also remove OMI pixels influenced by the OMI row anomaly (Torres et al., 2018), so there is reduced OMI data coverage over the record. However, we find this has minimal impact on our results with substantial proportions of data (e.g. millions of retrievals per year at the start and end of the OMI record) available for analysis in our study.".

New reference:

Torres, O., Bhartia, P.K., Jethva, H. and Ahn, C. 2018. Impact of the ozone monitoring instrument row anomaly on the long-term record of aerosol products. *Atmospheric Measurement Techniques*, **11**, 2701-2715, doi: 10.5194/amt-11-2701-2018.

Minor Comments:

Abstract, Line 35 "While GOME-2…" would sound better as "However, GOME-2…"

We have changed this accordingly.

Line 44 Here and throughout, references to IPCC AR5 should be updated with References to IPCC AR6. For this work, the relevant chapters of IPCC AR6 WG-I are Chapters 2 (Gulev et al., 2021), 6 (Szopa et al., 2021) and 7 (Forster et al., 2021). For example, the latest IPCC estimate of ozone's effective radiative forcing (ERF) is +0.47 [0.24 to 0.70] W m-2 (1750-2019, tropospheric + stratospheric ozone) (Forster et al., 2021).

We have updated the reference for the IPCC ozone radiative forcing range accordingly on Page 2 Line 49-50.